# Seasonal land ice-flow variability in the Antarctic Peninsula

Karla Boxall[1], Frazer D. W. Christie[1], Ian C. Willis[1], Jan Wuite[2] and Thomas Nagler[2]

[1]Scott Polar Research Institute, University of Cambridge, Cambridge, United Kingdom
[2]ENVEO IT GmbH, Innsbruck, Austria

*Correspondence to*: Karla Boxall (kb621@cam.ac.uk)

## Abstract

Recent satellite-remote sensing studies have documented the multi-decadal acceleration of the Antarctic Ice Sheet in response to rapid rates of ice-sheet retreat and thinning. Unlike the Greenland Ice Sheet, where historical, high temporal resolution satellite and in situ observations have revealed distinct changes in land ice flow within intra-annual timescales, observations

of similar seasonal signals are limited in Antarctica. Here, we use high spatial and temporal resolution Copernicus Sentinel-1A/B synthetic aperture radar observations acquired between 2014 and 2020 to provide the first evidence for seasonal flow variability of the land ice feeding George VI Ice Shelf (GVIIS), Antarctic Peninsula. Our observations reveal a distinct austral summertime (December – February) speedup of ~0.06 ± 0.005 m d$^{-1}$ (~22 ± 1.8 m yr$^{-1}$) at, and immediately inland of, the grounding line of the glaciers nourishing the ice shelf, which constitutes a mean acceleration of ~15% relative to baseline

(timeseries-averaged) rates of flow. These findings are corroborated by independent, optically derived velocity observations obtained from Landsat 8 imagery. Both surface and oceanic forcing mechanisms are outlined as potential controls on this seasonality. Ultimately, our findings imply that similar surface and/or ocean forcing mechanisms may be driving seasonal accelerations at the grounding lines of other vulnerable outlet glaciers around Antarctica. Assessing the degree of seasonal ice-flow variability at such locations is important for quantifying accurately Antarctica's future contribution to global sea-level

rise.

## 1.  Introduction

Three decades of routine Earth observation have revealed the progressive decay of the Antarctic Ice Sheet, evinced by accelerated rates of ice thinning, retreat and flow (Gardner et al., 2018; Konrad et al., 2018; The IMBIE Team, 2018; Rignot et al., 2019). This phenomenon has been ascribed to an array of atmospheric and oceanic forcing mechanisms impinging upon

the continent (Rignot et al., 2004; Thoma et al., 2008; Cook and Vaughan, 2010; Joughin et al., 2012a; Steig et al., 2012; Dutrieux et al., 2014; Paolo et al., 2018), from which resulting land ice losses are estimated to have totalled an average of ~109 ± 59 gigatons per year since 1992 (The IMBIE Team, 2018). Alongside satellite altimetry and gravimetry based assessments of ice-mass change, this trend has partly been constrained from satellite-derived velocity measurements acquired sporadically throughout the year (Rignot et al., 2011a; Mouginot et al., 2012), under the implicit (and unverified) assumption that no

discernible intra-annual (i.e., seasonal or shorter) variability in ice-flow exists (cf. Greene et al., 2018). In terms of intra-annual ice-flow variability, an overall dearth of systematic, high temporal resolution observations has also limited the ability to examine for such changes across Antarctica; this is in contrast to mid-latitude and Arctic ice masses, where the timing and large magnitude of seasonal ice-flow variability is now well observed (Iken et al, 1983; Hooke et al, 1989; Zwally et al., 2002; Moon et al., 2014; Kraaijenbrink et al., 2016; King et al., 2018). Within the context of recent ice-sheet modelling and mass

balance exercises (The IMBIE Team, 2018; Seroussi et al., 2020; Edwards et al., 2021), knowledge of any intra-annual variations in ice flow is critical for elucidating the processes controlling Antarctica's evolution in a changing climate.

In this study, we find evidence of seasonal ice-flow variability across the glaciers feeding the George VI Ice Shelf, Antarctic Peninsula. We use 6/12-day repeat-pass Copernicus Sentinel-1A/B synthetic aperture radar imagery for this purpose, together

with independent, 16-day repeat-pass observations acquired by the Landsat 8 Operational Land Imager. We then evaluate the potential mechanisms responsible for driving the observed seasonal ice-flow signals upstream of George VI Ice Shelf.

## 2.   George VI Ice Shelf

In this study, we investigate seasonal ice-flow variability across 21 glaciers feeding the glaciologically compressive George VI Ice Shelf (GVIIS) (Fig. 1). After Larsen C Ice Shelf, GVIIS is the second largest of the remaining ice shelves fringing the

Antarctic Peninsula (Holt et al., 2013), and has an areal extent of ~23,500 km$^2$. Ice-shelf flow bifurcates and advects towards both its northern (Marguerite Bay) and southern (Ronne Entrance) ice fronts at an average rate of 0.7 m d$^{-1}$ (255 m yr$^{-1}$), with flow averaging 0.08 m d$^{-1}$ (30 m yr$^{-1}$) and 1.1 m d$^{-1}$ (400 m yr$^{-1}$) along its Alexander Island and Palmer Land margins, respectively (Fig. 1). The thickness of the ice shelf ranges between approximately 100 and 600 m (Morlighem et al., 2020).

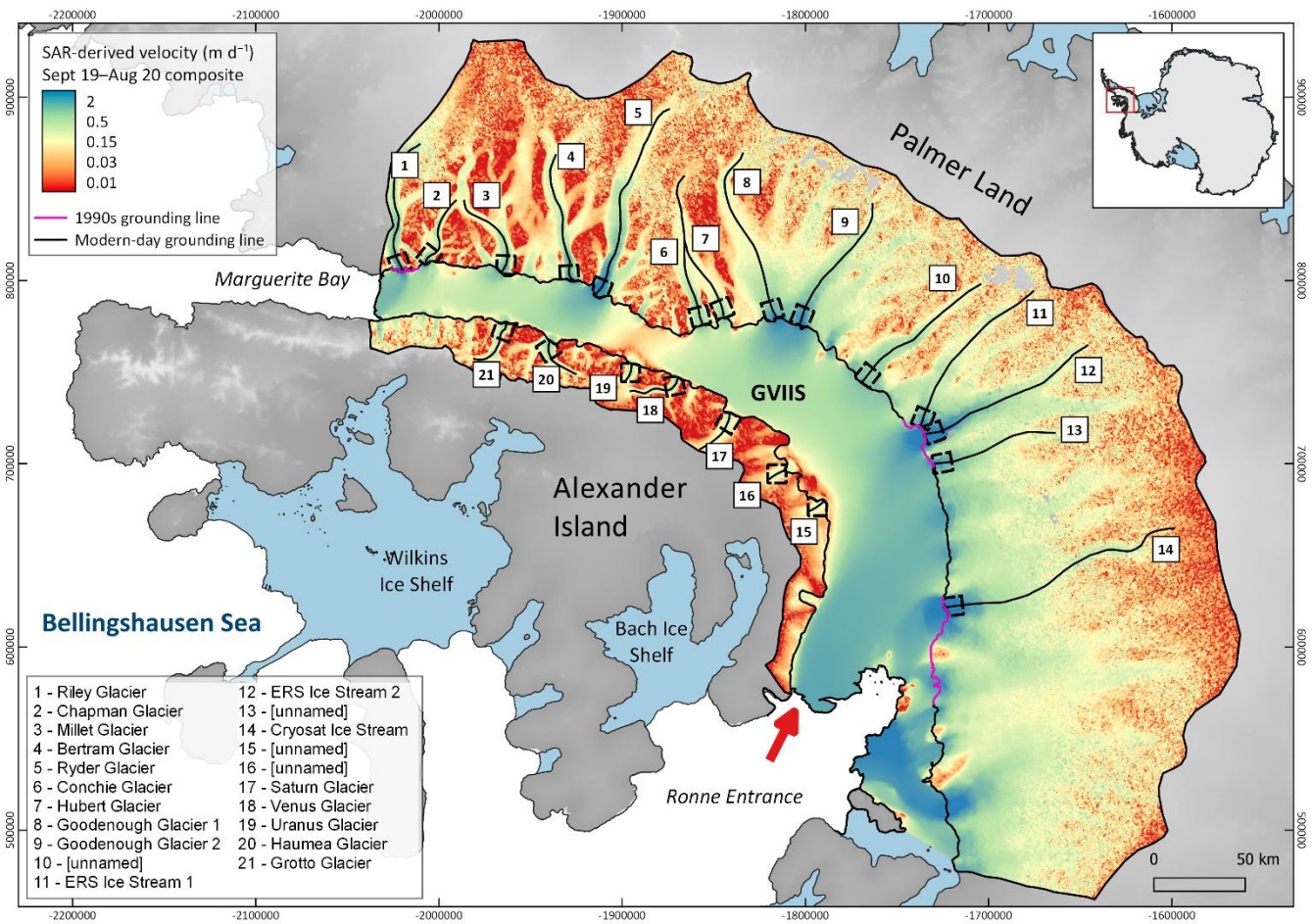

**Figure 1 – Mean ice flow of George VI Ice Shelf (GVIIS) and its drainage basins derived from Sentinel-1A/B synthetic aperture radar-derived observations acquired between 2019 and 2020. Where data coverage exists, the landward margins of GVIIS are delineated using the modern-day (2018-2020) grounding line (black) and, elsewhere, its position as imaged in the 1990s (pink; cf. Sect. 3.1). Drainage basin limits are from Mouginot et al. (2017). Background DEM is the Reference Elevation Model of Antarctica (Howat et al., 2019). Numbered flowlines delimit the centreline of the fastest-flowing outlet glaciers draining to GVIIS (named according to the UK Antarctic Place-names Committee), and the 10 km² dashed boxes located inland of the grounding line of these glaciers indicate the averaging regions used in the production of Figs. 4, 5, A1 and D1. Map projection: EPSG:3031. Inset map shows location. Arrow indicates the region of dominant CDW inflow as inferred from in situ ocean observations (Jenkins and Jacobs, 2008; see Sect. 5.2 for further discussion).**

Elsewhere in the Antarctic Peninsula, historical satellite observations have documented the abrupt, climate-driven disintegration of several ice shelves and the consequent dynamic acceleration of upstream glacial ice (Rott et al., 1996; Rack and Rott, 2004; Rignot et al., 2004; Scambos et al., 2004); this acceleration has increased Antarctica's net contribution to global sea-level (Scambos et al., 2004; The IMBIE Team, 2018). These events have been attributed primarily to the surface warming-induced presence of supraglacial meltwater lakes (Dirscherl et al., 2021), which are surmised to have instigated a process of rapid ice-shelf hydrofracture and collapse such as that observed most recently at Wilkins Ice Shelf (Fig. 1; cf.

Scambos et al., 2000; 2009; Banwell et al., 2013; Leeson et al., 2020). These phenomena have, in turn, been linked to an Antarctic Peninsula-wide increase in surface temperatures over most of the observational record (Vaughan et al., 2003). At GVIIS, intense supraglacial meltwater presence has been observed over the ice shelf since routine satellite observations began (Kingslake et al., 2017; Bell et al., 2018; Banwell et al., 2021), and melt season duration is one of the longest on the continent. This has led to the identification of GVIIS as a potential site for future ice-shelf disintegration (Holt and Glasser, 2022); recent modelling studies suggest that resultant land ice losses associated with such an event would contribute an ~8 mm rise in global sea-level by 2100 (Schannwell et al., 2018).

In addition to the effects of supraglacial meltwater, ocean-driven basal melting has had, and will likely continue to have, an important role in the evolution of GVIIS and its upstream glaciers (Pritchard et al., 2009; Holt et al., 2013; Paolo et al., 2015; Naughten et al., 2018). This is due to the flooding of relatively warm (~1-2 °C), high-salinity (~34.7 PSU) circumpolar deep water (CDW) beneath and offshore from GVIIS' sub-ice shelf cavity (Jenkins et al., 2010; Jenkins and Jacobs, 2008). Similar to the high rates of ice-shelf melting observed across the Amundsen and western Bellingshausen sectors in the past (Pritchard et al., 2012; Rignot et al., 2013), this CDW presence has driven basal melting of up to 7 m yr$^{-1}$ at GVIIS over the satellite era (Paolo et al., 2015; Adusumilli et al., 2018), and has been implicated as the primary mechanism responsible for the multi-decadal acceleration of GVIIS' fastest-flowing feeder glaciers (Hogg et al., 2017; Gardner et al., 2018; Winter et al., 2020).

## 3. Data and Methods

We use high spatial-temporal resolution, all-weather, day/night imaging Copernicus Sentinel-1A/B synthetic aperture radar (SAR) observations as our primary data source to both survey the seaward extent of the outlet glaciers feeding GVIIS and examine for seasonal variability in their flow. The methods used for these purposes are detailed below.

### 3.1 Grounding line delineation

Grounding lines are sensitive indictors of climate change and represent the boundary between seaward-flowing, terrestrial ice and adjoining, floating ice shelves. In parts of West Antarctica — including the Amundsen and western Bellingshausen sectors, especially — grounding lines have retreated pervasively in response to ocean forcing over the past ~50 years (Park et al., 2013; Rignot et al., 2014; Christie et al., 2016; 2018; Konrad et al., 2018). In comparison to the detailed knowledge of grounding-line migration within these sectors, however, there have been no high-resolution, spatially complete grounding line surveys at GVIIS since the mid-1990s (Rignot et al., 2016). Accurate and updated knowledge of GVIIS' grounding line is therefore of paramount importance for distinguishing precisely between grounded and floating ice.

To recover the location of GVIIS' modern-day grounding line, we employed double-differential interferometric SAR (DInSAR) processing techniques to all consecutive 6-day repeat-pass Sentinel-1A/B Interferometric Wide (IW) single look

complex (SLC) images acquired during extended austral wintertime (May-October) 2020. Extended austral wintertime imagery was used to maximise phase coherence between successive image pairs which may be degraded due to the attenuation

of radar waves by summertime supraglacial water presence (cf. Sect. 2). Similar to earlier work (Park et al., 2013; Rignot et al., 2014; Christie et al., 2016; 2022), we co-registered each successive image pair and removed the topographical component of phase from all subsequently generated interferograms, using the Reference Elevation Model of Antarctica (REMA; Howat et al., 2019). Assuming ice creep to be common between each SAR image, we then differenced all successive interferograms to locate the limit of tidally induced vertical ice-shelf flexure. This limit is represented as the landward extent of a band of

closely spaced fringes on double-differenced interferograms (Rignot et al., 2011b), and is an accurate proxy for the true grounding line which cannot be recovered directly by satellite-based imaging techniques (Fricker et al., 2009; Friedl et al., 2020). Finally, all interferograms were geocoded to EPSG:3031 (Antarctic Polar Stereographic projection) using REMA.

To supplement our grounding line observations across regions of poor phase coherence during extended wintertime 2020, we

filled data gaps using recently generated (albeit spatially discontinuous) Sentinel-1-derived grounding line information from 2018 (Mohajerani et al., 2021). Where this was not possible, for example in areas of phase aliasing resulting from very fast ice flow exceeding $\gtrsim$ 1.6 m d$^{-1}$, we used the 1994-1996 MEaSUREs grounding line dataset (Rignot et al., 2016; pink grounding lines in Fig. 1).

### 3.2 Derivation of ice velocity

Land ice velocities upstream of GVIIS were retrieved from all successive Sentinel-1 IW SLC image pairs acquired between October 2014 and August 2020 using a combination of coherent and incoherent offset tracking techniques as described in Nagler et al. (2015, 2021) and Wuite et al. (2015). These 12-day repeat-pass image pairs, which reduced to 6-day repeat following the launch of Sentinel-1B in April 2016, were then used to produce monthly composites of mean ice flow and associated grids of uncertainty (1σ) and valid pixel count (the number of non-NaN observations used in the production of each

monthly estimate). If during composite creation an image pair crossed into the neighbouring month between the reference and secondary images used to retrieve ice velocity, then it was weighted accordingly (cf. Nagler et al., 2015; Wuite et al., 2015). Final grids of monthly ice velocity, uncertainty and pixel count were outputted with a posting of 200 m and, like our double-difference interferograms, were geocoded from radar coordinates to EPSG:3031 (Antarctic Polar Stereographic) projection (cf. Section 3.1). Using our updated GVIIS grounding line product (cf. Sect. 3.1 and Fig. 1), floating ice was subsequently

masked from all monthly velocity grids. Proximal to the grounding line, where uncertainties associated with offset tracking techniques are typically much smaller relative to inland areas characterised by steep terrain and/or slow-flowing ice (Mouginot et al., 2012), the mean standard error associated with our grids totals 0.005 m d$^{-1}$ (1.8 m yr$^{-1}$). This value is comparable to that of other SAR-derived velocity products (Rignot et al., 2017; Friedl et al., 2021), and was calculated according to:

$$SE = \frac{\sigma}{\sqrt{n}} \qquad\qquad\qquad\qquad (1)$$

Where $SE$ denotes standard error, and $\sigma$ and $n$ denote the standard deviation and valid pixel count, respectively.

We averaged velocities over monthly timescales to minimise contamination associated with ionospheric and tropospheric delay between successive (6/12-day repeat pass) Sentinel-1 image acquisitions (Rosen et al., 2000; Selley et al., 2021). Moreover, across the Antarctic Peninsula, Sentinel-1A/B acquisitions are currently acquired in descending mode only (ESA, 2022), meaning that the velocity data utilised in this study represent the relative displacement of point targets from a single look angle only. Recent work has shown that below sub-monthly resolution, these velocities can be subject to bias owing to radar penetration differences associated with the freezing state of the snow-firn-ice interface between image acquisitions (Rott et al., 2020). This phenomenon can induce shifts in the radar line-of-sight (LOS) distance to target by up to several meters (Joughin et al., 2018; Rott et al., 2020), resulting in either an under- or overestimation of velocities depending on the flow direction of the ice relative to LOS (Rott et al., 2020). At present, reliable, sub-monthly estimates of the magnitude of this bias over the western Antarctic Peninsula are difficult to model owing to a lack of detailed in situ information on the temporally variable composition of the firn layer; for these reasons, monthly composites were also utilised to dampen the effect of this bias.

### 3.3 Detection of seasonal ice-flow variability

We examined for seasonal variations in ice flow using the methodology summarised in Fig. 2. First, we examined for spatial patterns of seasonal flow variability in our SAR-derived velocity datasets. We then supplemented our findings with intra-annual timeseries analyses at and proximal to the grounding line and corroborated observed patterns of ice-flow variation using independent satellite-derived observations. The methods associated with these analyses are discussed in Sects. 3.3.1, 3.3.2 and 3.3.3, respectively.

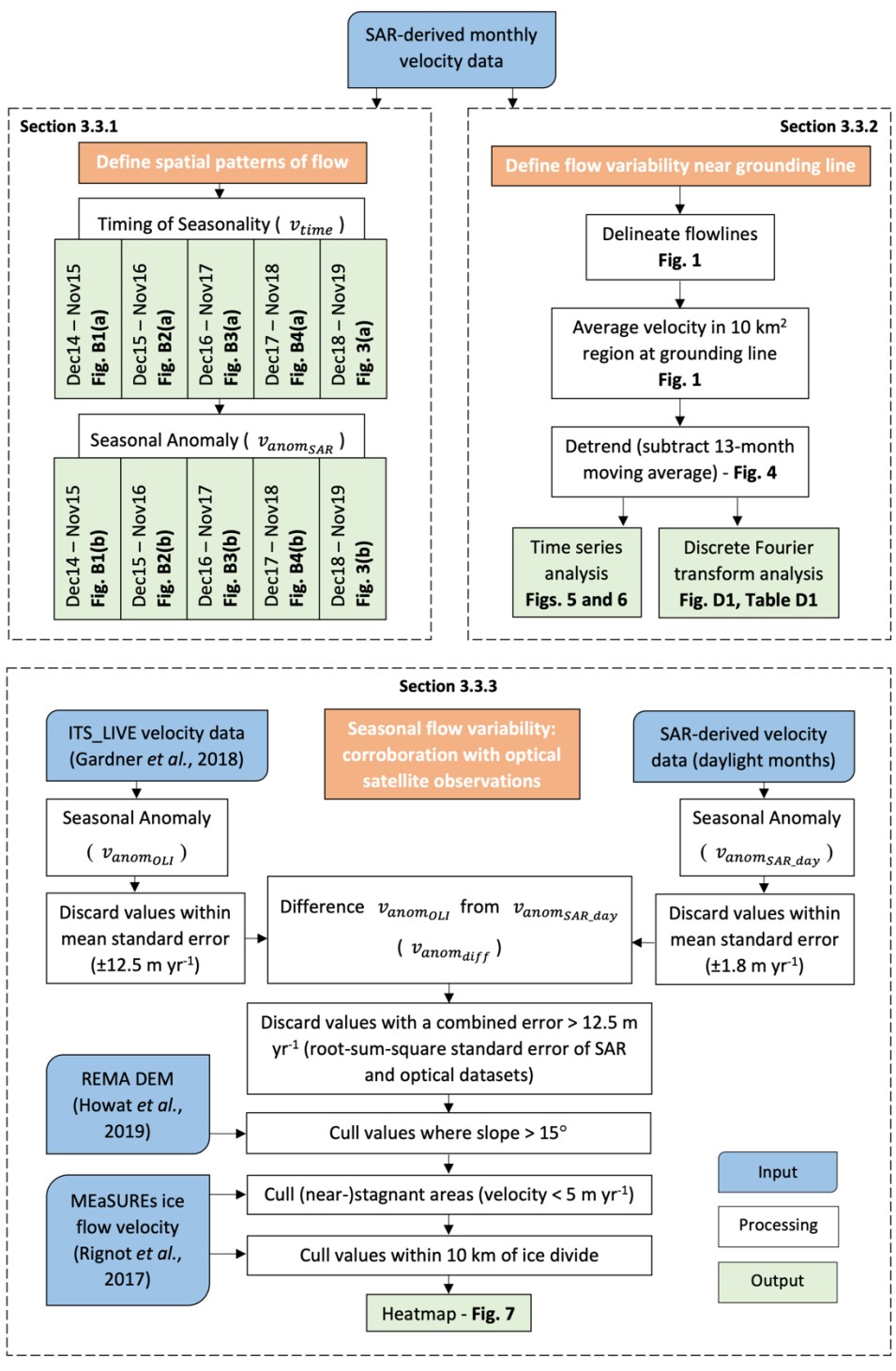

**Figure 2 - Workflow detailing the establishment of spatial patterns of flow (Sect. 3.3.1), the assessment of flow variability near the grounding line (Sect. 3.3.2) and the corroboration with optical satellite observations (Sect. 3.3.3).**

### 3.3.1 Spatial patterns of flow

Following Fig. 2, for each year spanning 2014 – 2019, we first constrained the month in which observed velocities were greatest on a pixelwise basis throughout the entire domain, $v_{time}$. This step was carried out to determine a) if a spatially coherent signal of ice-flow change was present upstream of GVIIS and, b) the season in which such a phenomenon occurred. Next, for each year (considered here to span December – November i.e., four complete seasons), the relative magnitude of any observed flow change was quantified in the form of a 'seasonal anomaly', $v_{anom_{SAR}}$. This metric is defined as the difference

in median velocity for the season (either DJF, MAM, JJA or SON) in which $v_{time}$ occurred, and the median across all other seasons that year. In the case of an austral summertime speedup associated with a maximum velocity observation in either December, January or February, for example, $v_{anom_{SAR}}$ is given by:

$$v_{anom_{SAR}} = (v_{12}, \widetilde{v_1, v_2}) - (v_3, \widetilde{\dots, v_{11}}) \qquad (2)$$


Where $v_n$ denotes the velocity magnitude as observed during month $n$ ($v_1$, January; …; $v_{12}$, December) Using this notation, positive $v_{anom_{SAR}}$ indicates a greater median velocity during austral summertime compared with all other seasons. In our calculation of $v_{time}$ and $v_{anom_{SAR}}$ above, pixels without continuous monthly data coverage throughout the entire year were culled from our analyses. Pixels falling within standard error bounds (cf. Sect. 3.2) were also discarded. Notably, the

calculation of either metric was not possible for 2019/20 given the lack of velocity data beyond August 2020 (Sect. 3.2).

### 3.3.2 Flow variability near the grounding line of GVIIS' fast-flowing outlet glaciers

To supplement our spatially resolved observations discussed above, we examined in greater detail the temporal variations in ice-flow near the coast of the outlet glaciers shown in Fig. 1. To do this, we calculated mean velocity and mean standard error (cf. Sect. 3.2) at monthly intervals within a 10 km$^2$ region located directly upstream of the grounding line between 2014 and

2020. Glaciers without sufficient modern-day grounding line coverage were excluded from further analysis (Fig. 1; cf. Sect. 3.1). Next, to accentuate intra-annual trends, we removed long-term trends in velocity over each 10 km$^2$ region (cf. Hogg et al., 2017; Gardner et al., 2018) using a 13-month moving average. At flowlines where the removal of a 13-month moving average was not possible, for example due to gaps in the time series, the data were excluded from our subsequent calculation of detrended monthly means. Finally, we quantified the dominant frequency of these trends using discrete Fourier transform

analysis. In signal processing, this technique decomposes a time domain function into the frequency domain, from which its most dominant frequency (i.e., number of cycles per unit of time) and phase and amplitude characteristics can be identified (Bracewell, 1978). To enable direct quantitative comparison between outlet glaciers, we fitted a cosine wave optimised to the most dominant frequency observed across all outlet glacier timeseries and extracted their associated phase and amplitude, which in this case denotes the approximate timing and magnitude of the seasonal velocity signals, respectively.

### 3.3.3 Seasonal flow variability: corroboration with optically derived satellite observations

To supplement our SAR-based observations, we also calculated a similar seasonal metric, $v_{anom_{OLI}}$, to that described in Sect. 3.3.1 using independent, spatially-temporally collocated velocity information derived from Landsat 8 Operational Land Imager (OLI) imagery acquired between 2014 and 2019 (Fig. 2). While Landsat 8's sun-synchronous orbit precludes any imaging outside extended austral summertime (September-April), derived velocity observations during this time offer an important dataset with which to corroborate any summertime ice-flow variability observed in the SAR record. This is due to the passive, on-nadir imaging characteristics associated with OLI which, unlike the SAR-based calculation of $v_{anom_{SAR}}$, are insensitive to any viewing geometry or snow/firn penetration-related biases (cf. Sect. 3.2). Any agreement between calculated values of $v_{anom_{SAR}}$ and $v_{anom_{OLI}}$ would therefore underscore the utility of our SAR-based methodology to detect extended summertime signals with confidence. Such agreement would, by extension, also imply a high degree of certainty in our ability to quantify ice-flow variability during non-daylight seasons.

We calculated $v_{anom_{OLI}}$ using Landsat 8-derived velocity grids acquired from the Inter-mission Time Series of Land Ice Velocity and Elevation (ITS_LIVE) data archive (Gardner et al., 2019). These grids were obtained for each unique Landsat granule over GVIIS and its environs, and represent velocities as constrained from feature tracking of all successive 16-day repeat-pass Landsat 8 image pairs acquired between January 2014 to April 2019 (Gardner et al., 2018; 2019). In total, 817 image pairs were utilised in this study which, comparable to our Sentinel-1-derived velocity grids, have a mean standard error of <0.034 m d$^{-1}$ (12.5 m yr$^{-1}$) (Gardner et al., 2018, 2019).

For a speedup during the austral summertime (DJF), for example, $v_{anom_{OLI}}$ was calculated using:

$$v_{anom_{OLI}} = \left( \widetilde{v2014_{12,1,2}, ...,} v2019_{12,1,2} \right) - \left( \widetilde{v2014_{3,4,9,10,11}, ...,} v2019_{3,4,9,10,11} \right) \qquad (3)$$

Where $v2014_{12,1,2}$ denotes the median velocity magnitude as observed during December, January and February 2014, respectively; $v2019_{12,1,2}$, same but for 2019 (i.e., the end of the observational record) and $v2019_{3,4,9,10,11}$, velocity magnitude spanning all non-summertime daylight months for 2019. Similar to Eq. (2), positive $v_{anom_{OLI}}$ would indicate a greater median velocity observed during austral summertime than all other (non--summertime) daylight months. Unlike in our derivation of $v_{anom_{SAR}}$, we calculated the median of all summer months over the entire 2014-2019 period to maximise spatial coverage at the expense of temporal resolution. This was due to the preponderance of clouds in the Landsat record and resulting lack of velocity coverage at monthly resolution. Prior to any further analysis, values of $v_{anom_{OLI}}$ falling within standard error (<0.034 m d$^{-1}$) were removed.

To corroborate our SAR-derived metric (Sect. 3.3.1), we differenced $v_{anom_{OLI}}$ from a second measure of $v_{anom_{SAR}}$ whose temporal limits were clipped to match the daylight months imaged by Landsat 8 (September – April; Fig. 2). Like $v_{anom_{OLI}}$, this second metric, $v_{anom_{SAR\_day}}$, was calculated using Eq. (3), and all values lying within SAR-derived standard error bounds

($\pm0.005$ m d$^{-1}$) were discarded during calculation. Upon differencing $v_{anom_{SAR\_day}}$ and $v_{anom_{OLI}}$, we filtered the resulting grid, $v_{anom_{diff}}$, to remove pixels containing unrealistically high values. Visual examination of the raw image data associated with these pixels (not shown) reveal such values to be associated with regions of frequent cloud cover in the Landsat 8 record, which result in more poorly refined velocity estimates with a high standard error. For this purpose, we discarded all instances of $v_{anom_{diff}}$ with a combined error surpassing 0.034 m d$^{-1}$ (12.5 m yr$^{-1}$), calculated from the mean standard errors associated

with our SAR- and optically derived datasets summed in quadrature. Finally, we culled all pixels located within 10 km of the ice divide (where signal-to-noise is often poor using offset tracking techniques; Mouginot et al. (2012)), across regions of complex topography (>15° slope) and near-stagnant (<0.013 m d$^{-1}$) flow as defined by, respectively, REMA DEM (Howat et al., 2019) and an independent velocity dataset (Rignot et al., 2017) (Fig. 2). All remaining valid pixels were then used to produce a heatmap emphasising the spatial coherence of seasonal ice-flow speedup observed in both our SAR- and optically

derived velocity records.

## 4. Results

### 4.1 GVIIS' grounding line

Our new 2018-2020 grounding line location compilation (cf. Sect. 3.1) provides an important update on GVIIS' geometry, whose only other publicly available InSAR-based records were derived from observations acquired in the mid-1990s (Rignot

et al., 2016). In total, our compilation provides revised grounding line information across 76% of GVIIS' coastal margin (Fig. 1). Since the mid-1990s, however, we detect no significant change in grounding line location, with the position of the mid-1990s grounding line falling firmly within the range of tidally induced grounding line locations observed by Sentinel-1A/B (1-5 km depending on the glacier; Fig. B1). In the following sections, all results presented are derived from observations located inland of the high-tide (i.e., most landward) 2018-2020 grounding line location.

### 4.2 Seasonal ice-flow variability

### 4.2.1 Spatially resolved patterns of flow

Figures 3 and C1-4 show clear seasonal ice-flow variability along the entire GVIIS coastal margin, with the greatest magnitude of seasonality clustered tightly at or proximal to the deep-bedded grounding lines of the fastest-flowing outlet glaciers (cf. Fig. 1). Along both the Alexander Island and Palmer Land coasts, our calculation of $v_{time}$ (i.e., the month of maximum velocity; cf. Sect. 3.3.1) reveals the ubiquitous occurrence of ice-flow speedup during austral summertime months (DJF; Figs. 3(a) and

C1-4(a)). This phenomenon is mirrored in our calculation of $v_{anom_{SAR}}$ (Figs. 3(b) and C1-4(b)), which provides a quantitative measure of the observed ice-flow speedup. For 33% of the outlet glaciers numbered in Fig. 1, observed summertime velocities are ≥0.1 m d$^{-1}$ (36.5 m yr$^{-1}$) faster than the median velocity of all non-summertime seasons, and summertime velocity increases near the grounding line of all (100%) glaciers exceed error bounds (0.005 m d$^{-1}$ (1.8 m yr$^{-1}$)). Notably, no coherent seasonal

signal exists beyond ~10-20 km of the grounding line, where $v_{time}$ is randomly distributed across all 12 months and $v_{anom_{SAR}}$ shows no associated clustering.

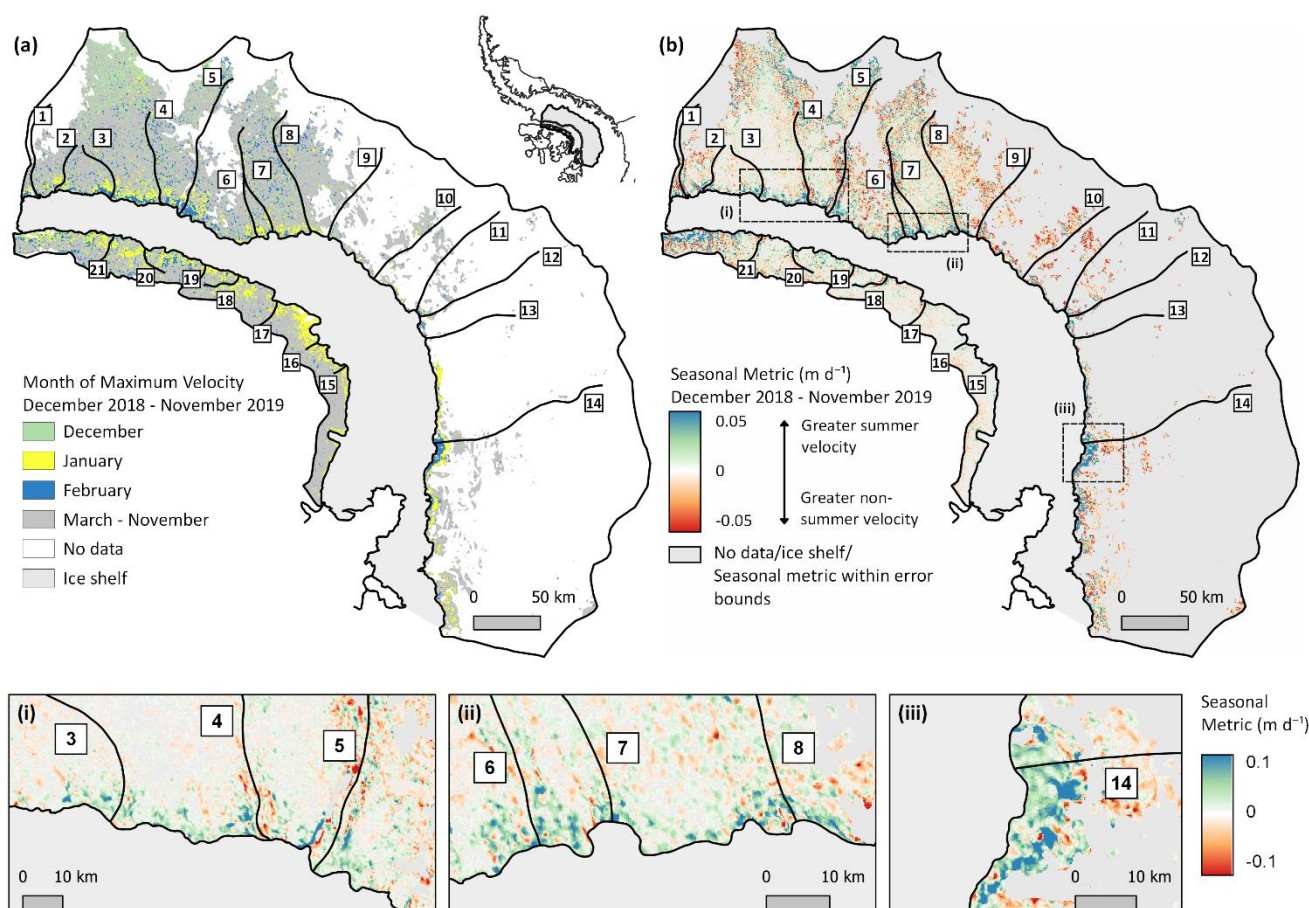

**Figure 3 – Spatially resolved patterns of seasonal ice-flow as observed between December 2018 and November 2019.** (*a*) shows the month of maximum velocity ($v_{time}$); (*b*) the calculated velocity anomaly ($v_{anom_{SAR}}$). In (*b*), positive velocity anomalies (blue) indicate summertime speed-up; negative (red), greater velocity during non-summertime months. Dashed boxes denote the location of the inset boxes (*i-iii*). See also Figs. C1-C4.

### 4.2.2 Temporal flow variability near the grounding line

Near the modern-day grounding line, spatially averaged observations of both raw (Fig. 4) and detrended (Fig. 5), monthly ice-flow between 2014 and 2020 are consistent with the seasonal signals discussed above (Figs. 3 and C1-4). There, 94% of the outlet glaciers underwent an average summertime (DJF) speedup of 0.06 m d$^{-1}$ (min. 0.02 m d$^{-1}$; max. 0.15 m d$^{-1}$; 1$\sigma$ = 0.037 m d$^{-1}$), or 21.9 m yr$^{-1}$ (Figs. 5 and A1). On average, this corresponds to a ~15% summertime increase in ice-flow relative to baseline (timeseries-averaged) rates along GVIIS' grounding line. Furthermore, our observations show that 88% of the glaciers underwent velocity minima (blue in Fig. 5) during non-summertime months, and the error bounds associated with all velocity minima fell firmly outside those corresponding to velocity maxima (red in Fig. 5). Of the velocity profiles shown in Fig. 5, the only glaciers opposing this general pattern were those corresponding to Flowlines 8 (Goodenough Glacier 1) and 10 (unnamed), which exhibit a bimodal and non-summertime speedup signal, respectively. The latter likely arises from poor temporal data coverage, while the double peak associated with Flowline 8 is attributed to an anomalous wintertime speedup in 2016 which dominated the mean velocity signal of this glacier.

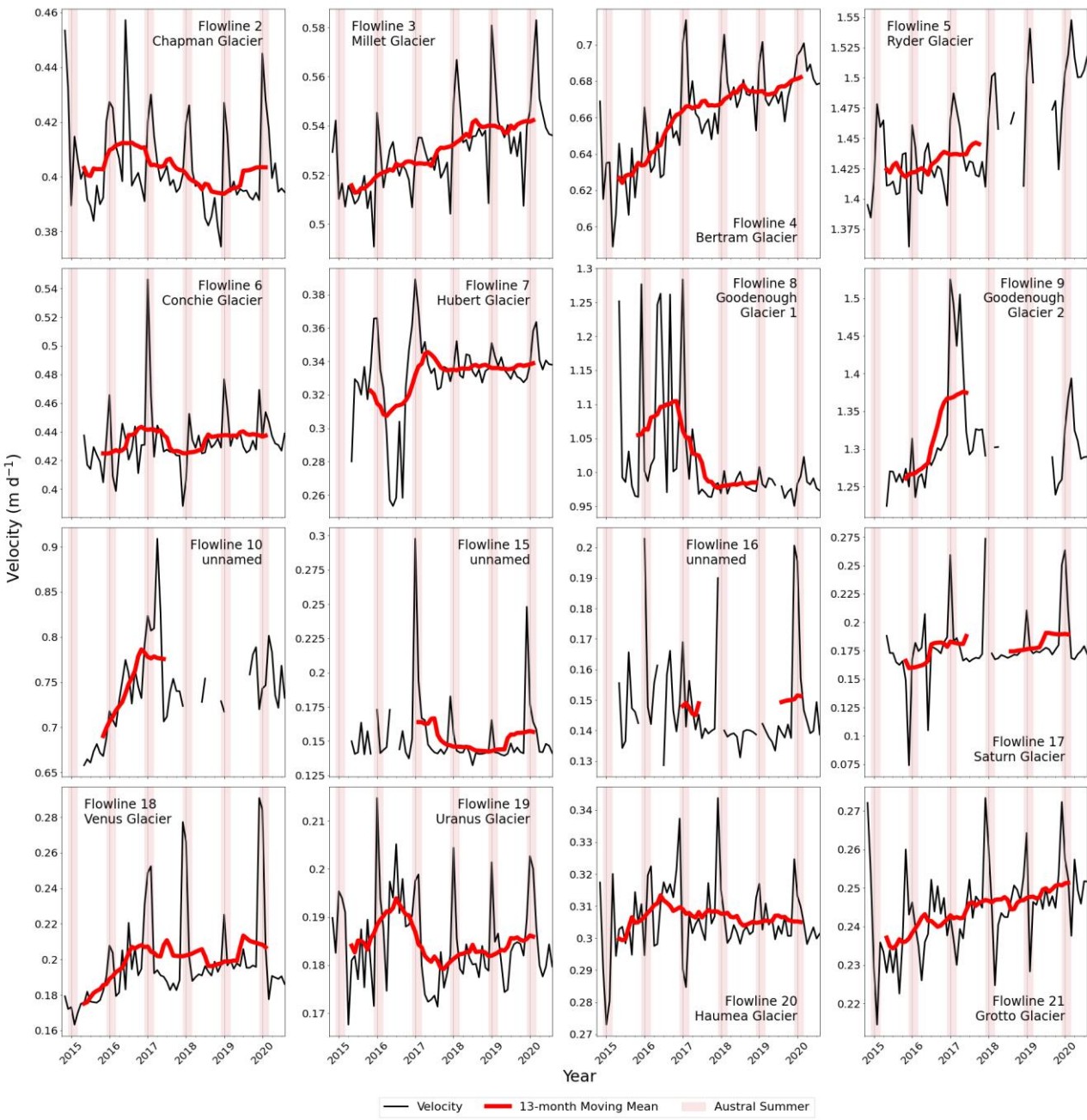

**Figure 4 - SAR-derived observations of ice flow for each month between October 2014 and August 2020 (black line) and the 13-month moving mean ice flow (red line). Observations are averaged across the 10 km² dashed boxes shown in Fig. 1. Translucent red shading denotes the timing of austral summertime (December–February, inclusive). Note that the y-axis limits vary between panels.**

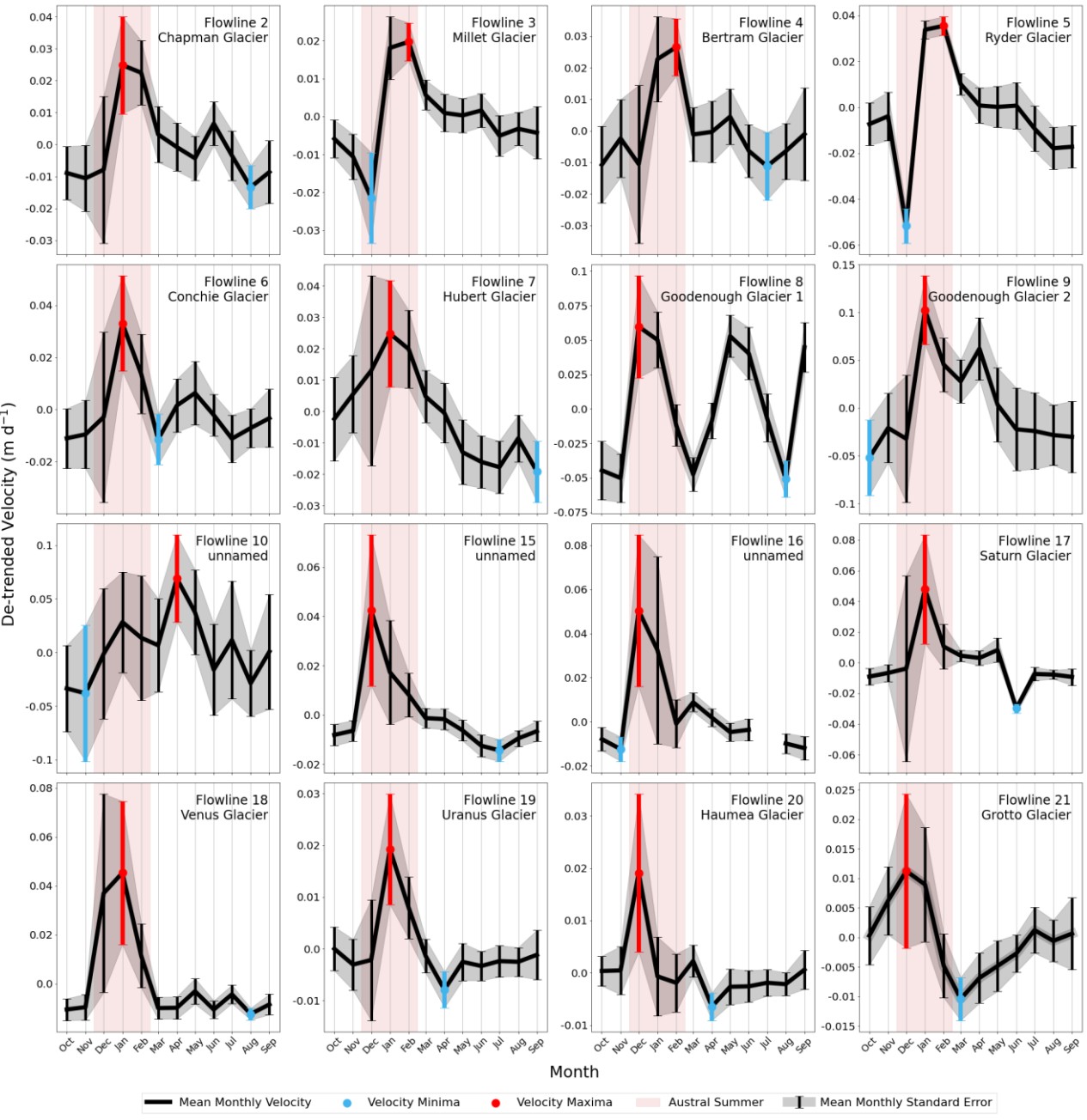

**Figure 5 – Detrended SAR-derived observations of mean monthly ice flow between 2014 and 2020 (black) and associated mean monthly standard error (grey shading). Observations are averaged across the 10 km² dashed boxes shown in Fig. 1. Blue and red error bounds denote velocity minima and maxima, respectively. Translucent red shading denotes the timing of austral summertime (December-February, inclusive). Note that the y-axis limits vary between panels.**

Spatially, our results are also suggestive of an apparent regional contrast in the timing of summertime speedup, whereby the glaciers nourishing GVIIS from Alexander Island appear to have accelerated in early-to-mid summertime (December – January) compared to those draining from Palmer Land (mid-to-late summertime/autumn (January – April)) (Fig. 6). This finding is consistent with the approximate timing of peak velocity as determined from discrete Fourier transform analysis (Appendix D), which further reveals that most (88%) outlet glaciers have a dominant intra-annual signal characterised by a cosine wave with a frequency of 1 (implying one complete seasonal cycle per 12-month period; cf. Fig. D1). Within this trend, Figs. 4 and 5 also present evidence of the ability of select GVIIS' outlet glaciers to influence each other across the ice shelf, whereby the earlier acceleration of Alexander Island's glaciers initially arrest those flowing from Palmer Land, delaying the onset of their acceleration until the late summertime (compare, for example, the timing of peak velocity at the geographically opposite Flowlines 21 and 3, and Flowlines 20 and 4-5). Upon late-summertime acceleration, Palmer Land's glaciers then arrest the flow of those on Alexander Island in a similar manner. Continued monitoring of this phenomenon beyond the current Sentinel-1a/b observational record may shed additional light upon the importance of this mechanism (or otherwise) for controlling the seasonal signal exhibited at these outlet glaciers.

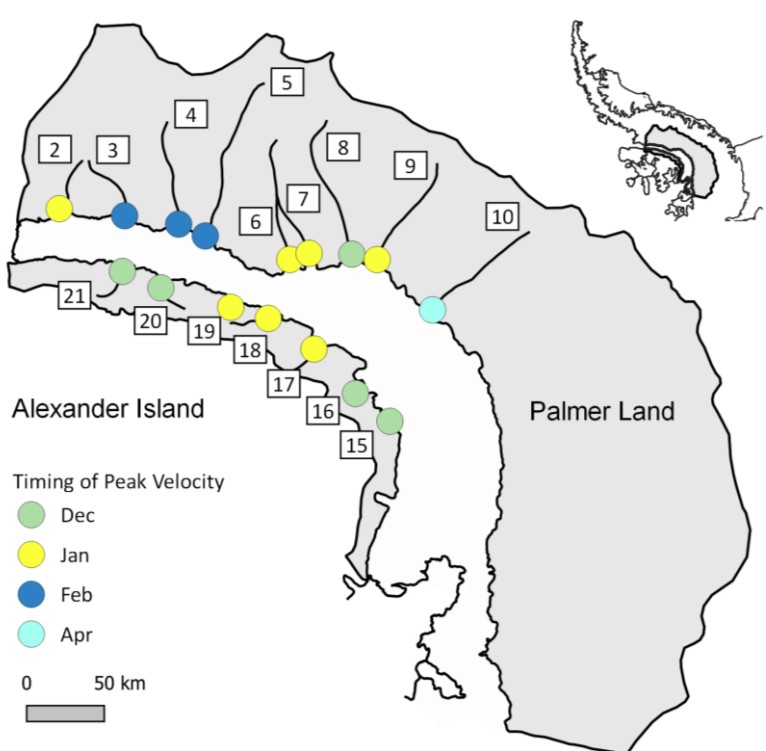

**Figure 6 - Month of maximum velocity obtained from the detrended SAR-derived observations shown in Fig. 5. Inset map shows the location of GVIIS' drainage basins.**

### 4.2.3 SAR vs. Optical observations of flow

Figure 7 reveals strong agreement between our SAR- and optically derived seasonal observations (i.e., $v_{anom_{SAR\_day}}$ and $v_{anom_{OLI}}$, cf. Sect. 3.3.3), whereby summertime speedups are observed in both datasets within ~10-20 kilometres of the grounding line. There, coincident Sentinel-1-/Landsat 8-derived speedup observations are tightly clustered and confined to the lateral dimensions of the fast-flowing outlet glaciers (contours in Fig. 7). Upon examination of the velocity field (Fig. 1) in conjunction with the phenomena observed in Fig. 7, we infer the clustered regions of agreement farther inland (~40-150+ km from the grounding line) to be falsely identified regions of speedup falling close to combined sensor error limits (cf. Sect. 3.3.3). There, clustering resides mostly over areas of near-stagnant flow unlikely to have experienced significant seasonal variability (~10 m yr$^{-1}$). In the few instances where such phenomena are located over regions of faster flow, the spatial distribution of clustering is not bound to the dimensions of the tributary glaciers shown in Figs. 1 and 7, and so is assumed also not to represent a true geophysical signal.

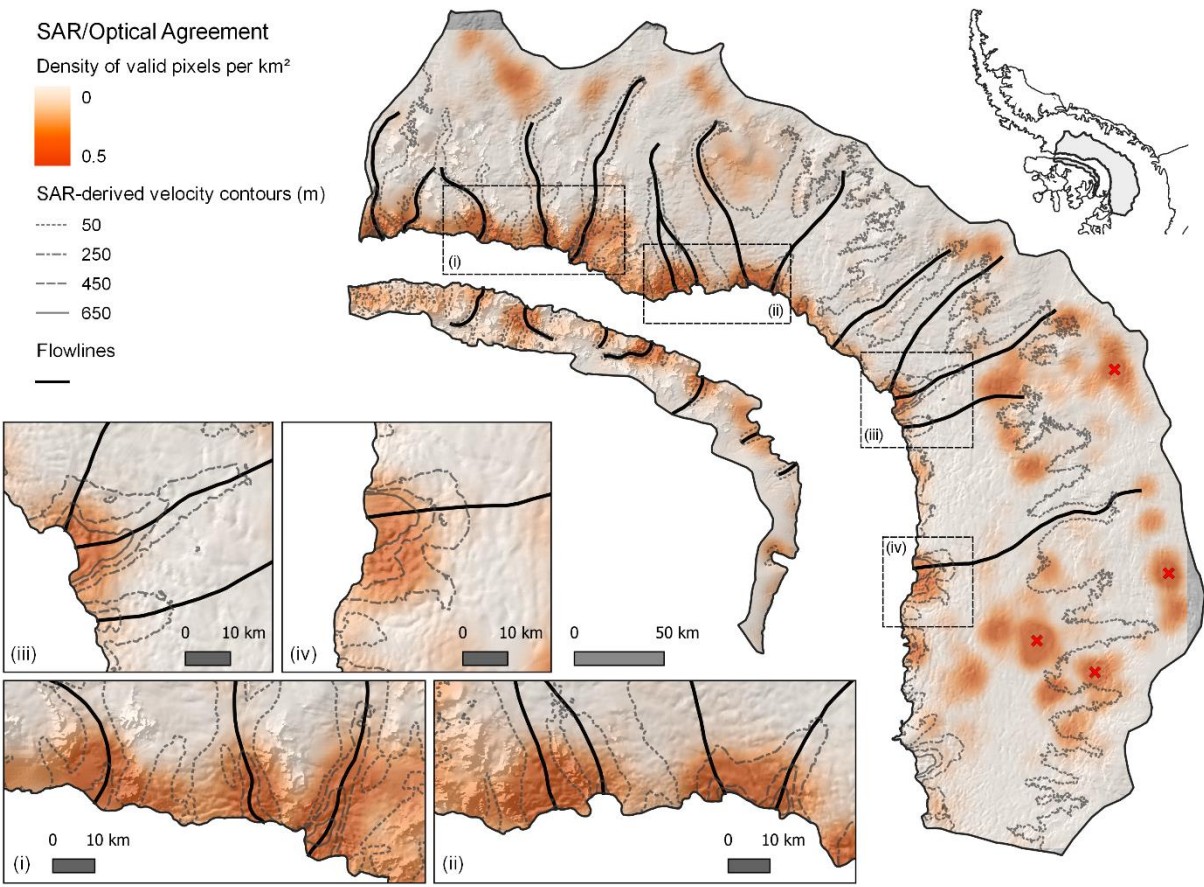


**Figure 7 - Heatmap indicating the spatial coherence of seasonal ice-flow variability observed by both SAR and optical imaging techniques. Warmer colours denote a higher density (per km²) of pixels having undergone speedup of comparable magnitude (±10 m yr⁻¹) during austral summertime, as observed in both our SAR- ($v_{anom_{SAR\_day}}$) and optically derived ($v_{anom_{OLI}}$) records (cf. Sect. 3.3.3). Contours are derived from our SAR-derived, timeseries-averaged velocity observations (Sect. 3.2). Background DEM is**
**derived from the Reference Elevation Model of Antarctica (Howat et al., 2019). Dashed boxes denote the location of the inset boxes (i-iv). Inset map shows the location of GVIIS' drainage basins. Red crosses represent examples of areas of clustering not representative of a true geophysical signal.**

## 5. Discussion

Our observations present unambiguous evidence for seasonal ice-flow variability on the grounded ice draining to GVIIS.
Proximal to the grounding line, glacier velocity increased during summertime months (December – February) by $0.06 \pm 0.005$ m d⁻¹ on average, constituting a ~15% speedup relative to baseline (timeseries-averaged) velocities. In the following sections, we evaluate the potential surface and ocean forcing mechanisms driving this phenomenon, which are important, ultimately, for better constraining the future timing and evolution of the Antarctic Ice Sheet's decay.

## 5.1 Surface forcing

The role of surface-sourced meltwater in stimulating seasonal accelerations in land ice flow is well established on valley glaciers and the Greenland Ice Sheet (Iken et al, 1983; Hooke et al, 1989; Zwally et al., 2002; Moon et al., 2014; Kraaijenbrink et al., 2016). At both locations, early summertime surface water inputs to a subglacial drainage system drive near-instantaneous reductions in effective pressure, increased basal sliding and the acceleration of the entire ice column (Iken, 1981; Schoof, 2010). Enabled primarily through the mass drainage of supraglacial meltwater via surface-visible 'moulins' (which are

themselves formed through the sustained meltwater-driven erosion of the ice column over one or more summers), such velocity accelerations are typically short-lived as the subglacial hydrological drainage system becomes more efficient through time, thereby increasing effective pressure and arresting flow (Bartholomew et al., 2010). Near the surface of the Greenland Ice Sheet, the drainage of perennial, summer meltwater-fuelled 'firn aquifers' may also deliver large quantities of meltwater to the bed via crevasses and/or other englacial pathways (Harper et al., 2012; Koenig et al., 2013), instigating similar, transient

accelerations in ice velocity (Schoof et al., 2010).

In Antarctica, the clear velocity signals we observe inland of GVIIS' grounding line (Figs. 4 and 5) emulate closely the summertime accelerations observed in valley glaciers and the Greenland Ice Sheet, implying that they may be driven by similar surface meltwater-related processes. The relatively short-lived (~1-2 months duration) velocity maxima followed by sharp

deceleration trends at most glaciers lend credence to this interpretation (Figs. 4, 5 and A1), with the latter resembling a 'Greenland-style' late- to post-summertime switch towards more efficient subglacial drainage. A current lack of in situ observations, however, makes this hypothesis difficult to verify. Nonetheless, we note further that for 75% of the outlet glaciers nourishing GVIIS, the greatest instances of ice-flow speedup occurred during either the austral summertime of 2016/17 or 2019/20 (Fig. A1), which correspond to years characterised by exceptional surface melting on the ice shelf (Banwell et al.,

350 2021).

Despite the seemingly close correspondence between surface-meltwater forcing and the seasonal signals we observe at GVIIS' glaciers, satellite observations show that supraglacial meltwater presence and persistence is limited inland of Antarctica's grounding zone (Dirscherl et al., 2021; Johnson et al., 2022), and no obvious regional contrasts in melt exists near the grounding

line of Alexander Island and Palmer Land (Trusel et al., 2013; Bell et al., 2018). At GVIIS, routine satellite observations have also revealed minor trends of decreasing meltwater presence over most of the 21st Century (Johnson et al., 2022), which is consistent with a previously documented, pervasive cooling of the Antarctic Peninsula from the late 1990s onwards (Turner et al., 2016; Adusumilli et al., 2018). Inland, we expect that melt rates will have similarly decreased but at a greater rate given the lapse rate associated with the Antarctic Peninsula's mountainous terrain. Together with the variable thicknesses of the

glaciers spanning GVIIS' perimeter (Morlighem et al., 2020), which would presumably require differing amounts of meltwater flux to form surface-to-bed moulins and enhance basal sliding, these findings suggest that rapid surface meltwater drainage

events alone are unlikely to explain the region-wide, year-by-year seasonal speedup signals we observe near the grounding line. Summertime coherence in both the interferograms used to locate the position of the grounding line in 2018 (cf. Section 3.1; Mohajerani et al., 2021) and our offset-tracking-derived velocity estimates (cf. Section 3.2), alongside a previously documented absence of any such rapid meltwater drainage events in the Antarctic Peninsula (Rott et al., 2020), further support this assertion.

At depth (i.e., beyond that detectable by microwave sensors, whose radiation penetrates typically a few metres into the firn column), Greenland-style firn aquifer-related drainage events could also be involved in driving the seasonal velocity signals we observe. Previous in situ campaigns have confirmed the presence of such aquifers on Wilkins Ice Shelf to the north (Fig. 1), and these have been implicated as potential drivers of Wilkins' past disintegration events (Montgomery et al., 2020). We note, however, that the formation and persistence of firn aquifers requires high levels of surface melt and accumulation (Harper et al., 2012; Koenig et al., 2012; Montgomery et al., 2020): phenomena which may not be prevalent inland of GVIIS during most of the Sentinel-1 era given the pervasive cooling of the Antarctic Peninsula over approximately the past two decades noted above (cf. Turner et al., 2016). Notwithstanding this cooling, long-term model outputs suggest that the spatial distribution of firn aquifers in the Antarctic Peninsula is limited largely to the northern reaches of Wilkins Ice Shelf, and that no aquifers reside inland of GVIIS (van Wessem et al., 2021) — patterns which follow closely the thermal limit of ice-sheet viability over this region of Antarctica (Cook & Vaughan, 2010). While we acknowledge that the model estimates of, for example, van Wessem et al. (2021) may underestimate firn aquifer presence, the rapidly steepening topography inland of GVIIS' grounding line (which often exceeds 1° of slope; Howat et al., 2019; cf. Fig. 1) is presumably also unconducive to their formation and persistence compared with the relatively flat Wilkins Ice Shelf.

**5.2 Ocean forcing**

The seasonal velocity signals we observe at and proximal to GVIIS' grounding line (Figs. 3, 4, 5 and A1) may also be diagnostic of seasonal fluctuations in ocean forcing. As discussed in Section 2, this interpretation is supported firstly by in situ oceanographic observations revealing the widespread inflow and flooding of relatively warm circumpolar deep water (CDW) to GVIIS' cavity, which is sourced from the continental shelf via a net northwards throughflow from Ronne Entrance (Jenkins and Jacobs, 2008). There, the strongest inflows of CDW have been observed to occur underneath its northern margin proximal to Alexander Island (red arrow in Fig. 1; after Jenkins and Jacobs, 2008) which may, by extension, explain the observed, earlier onset of summertime speedup at and proximal to the grounding line along that stretch of coastline relative to Palmer Land (Fig. 6). On the basis of these earlier observations, we further expect that enhanced CDW upwelling in the cavity, enabled by the buoyancy driven advection of ice-shelf meltwater entrained within the northerly throughflow and deflected towards Alexander Island due to Coriolis forcing, may have also maximised this contrasting regional melting effect. This hypothesis is consistent with in situ- and modelling-based estimates of GVIIS' sub-shelf circulation (Jenkins and Jacobs, 2008; Holland

et al., 2010) and, more broadly, with inferred patterns of melting observed recently along the Coriolis-favoured flank of Dotson Ice Shelf, (Gourmelen et al., 2017) and Getz Ice Shelf (Alley et al., 2016).

It is important to note, however, that the findings of Jenkins and Jacobs (2008) do not present any evidence for seasonality in CDW presence and/or depth owing to the limited timeframe in which these in situ observations were collected (less than two days' worth of continuous measurements in March 1994). Nonetheless, recent research has revealed two possible mechanisms through which sea ice conditions offshore from GVIIS may control CDW draft in its sub-shelf cavity. First, modelling experiments, emulating closely the observational records of Jenkins & Jacobs (2008), have suggested a process of wintertime sea-ice growth, brine rejection and resulting convection throughout the mixed layer that leads to a thickening of the underlying CDW (Holland et al., 2010; Petty et al., 2014). These processes drive a seasonal cycle in melt rate at GVIIS' grounding line (Holland et al., 2010), greatest around mid-to-late wintertime, which would precede the resulting summertime accelerations in ice flow we observe by ~3-5 months. Several ice-sheet modelling studies have suggested a lag time of several weeks to months from the onset of ocean-induced melt to surface ice acceleration at the grounding line (Vieli and Nick, 2011; Joughin et al., 2012b), suggesting that this timescale may be plausible. Second, in situ observations from the neighbouring Amundsen Sea Sector have revealed that sea-ice growth and associated brine rejection can alternatively result in a destratification of the water column and thus restrict CDW inflow during austral wintertime (Webber et al., 2017). This mechanism would, by implication, facilitate enhanced summertime melt at the grounding line more in-phase with the accelerations in ice flow we observe.

Ultimately, a dearth of oceanographic observations in the Bellingshausen Sea hinders our ability to ascertain which mechanism is the dominant control on CDW influx to GVIIS' sub-shelf cavity, justifying the future collection of detailed oceanographic data in this region. Such data would also yield high-resolution (and potentially more representative) insights into the nature of oceanic circulation beneath GVIIS. Indeed, we estimate that an ~8-16 cm s$^{-1}$ northwards throughflow of CDW would be required over the course of ~1-2 months to induce the relatively narrow summertime speedup windows observed along the entirety of GVIIS' 420 km long cavity (Figs. 4, 5 and A1), assuming laminar and linear flow. These rates are up to almost an order of magnitude greater than the observationally constrained estimates reported in Jenkins & Jacobs (2008; ~2.5 cm s$^{-1}$), although the latter, which were collected over ~30 hours, may not necessarily be representative of seasonally averaged rates of flow. Elsewhere in Antarctica, longer-term in situ observations have revealed much greater rates of sub-ice shelf CDW circulation (~8-20 cm s$^{-1}$; Jacobs et al., 2013; Jenkins et al., 2018), suggesting that similar speeds may, in fact, be plausible underneath GVIIS.

Finally, we note that beyond relatively local-scale ocean forcing, modelling experiments suggest that CDW influx to GVIIS' cavity is relatively insensitive to far-field intra-annual-to-decadal-scale atmosphere-ocean variability (Holland et al., 2010). This lies in contrast to the atmosphere-ocean processes controlling CDW transmission to the Amundsen and wider Bellingshausen Sea coastal margins (i.e., wind-driven Ekman-transport; cf. Steig et al., 2012; Dutrieux et al., 2014; Christie et

al., 2018; Jenkins et al., 2018; Paolo et al. 2018), further implicating relatively local-scale, sea-ice induced oceanographic modification as a key potential control on the seasonal ice velocity signals we observe at GVIIS.

## 6. Summary and Implications

We provide the first evidence for seasonal flow variability of land ice draining to George VI Ice Shelf (GVIIS). Using monthly Sentinel-1 SAR-based velocity information derived from high-frequency (6-/12-day) repeat-pass observations, we detect a ~15% mean austral summertime speedup of the outlet glaciers feeding GVIIS at, and immediately inland of, the grounding line. This seasonal variability is corroborated by independent, optically derived observations of ice flow. Both surface- and oceanic-forcing mechanisms are evaluated as potential controls on this seasonality, although insufficient observational evidence currently exists with which to verify the relative importance of each. Elucidating the precise surface and/or ocean mechanisms governing GVIIS' outlet glacier flow variability is therefore a critical area for future research.

Ultimately, our findings imply that other glaciers in Antarctica may be susceptible to — and/or currently undergoing — similar ice flow seasonality including the highly vulnerable and rapidly retreating Pine Island and Thwaites glaciers. We further expect that such behaviour would not necessarily be captured in, for example, input-output method mass balance calculations, leading to potentially misestimated rates of ice-sheet mass loss. Accurately ascertaining the nature of seasonal ice, ocean and/or atmospheric interactions at such locations is important, therefore, for better understanding, modelling, and ultimately refining projections of the rate at which future Antarctic ice-losses will contribute to global sea-level rise.

**Appendices**

**Appendix A**

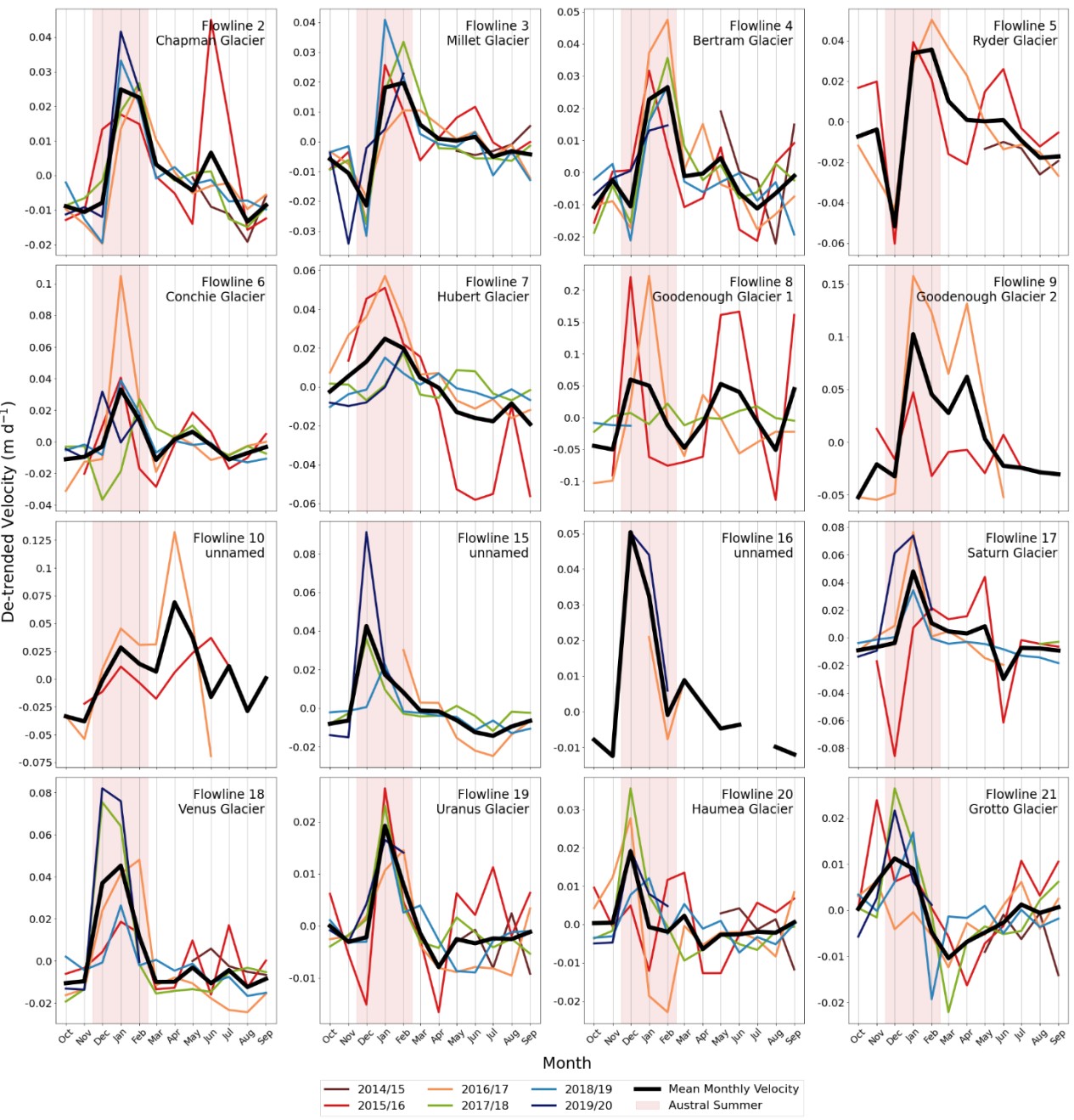

**Figure A1 - Detrended SAR-derived observations of mean monthly ice flow for each year between 2014 and 2020 (coloured lines) and the mean monthly ice flow calculated over the entire time series (black line; same as Fig. 5). Observations are averaged across the 10 km² dashed boxes shown in Fig. 1. Translucent red shading denotes the timing of austral summertime (December–February, inclusive). Note that the y-axis limits vary between panels.**

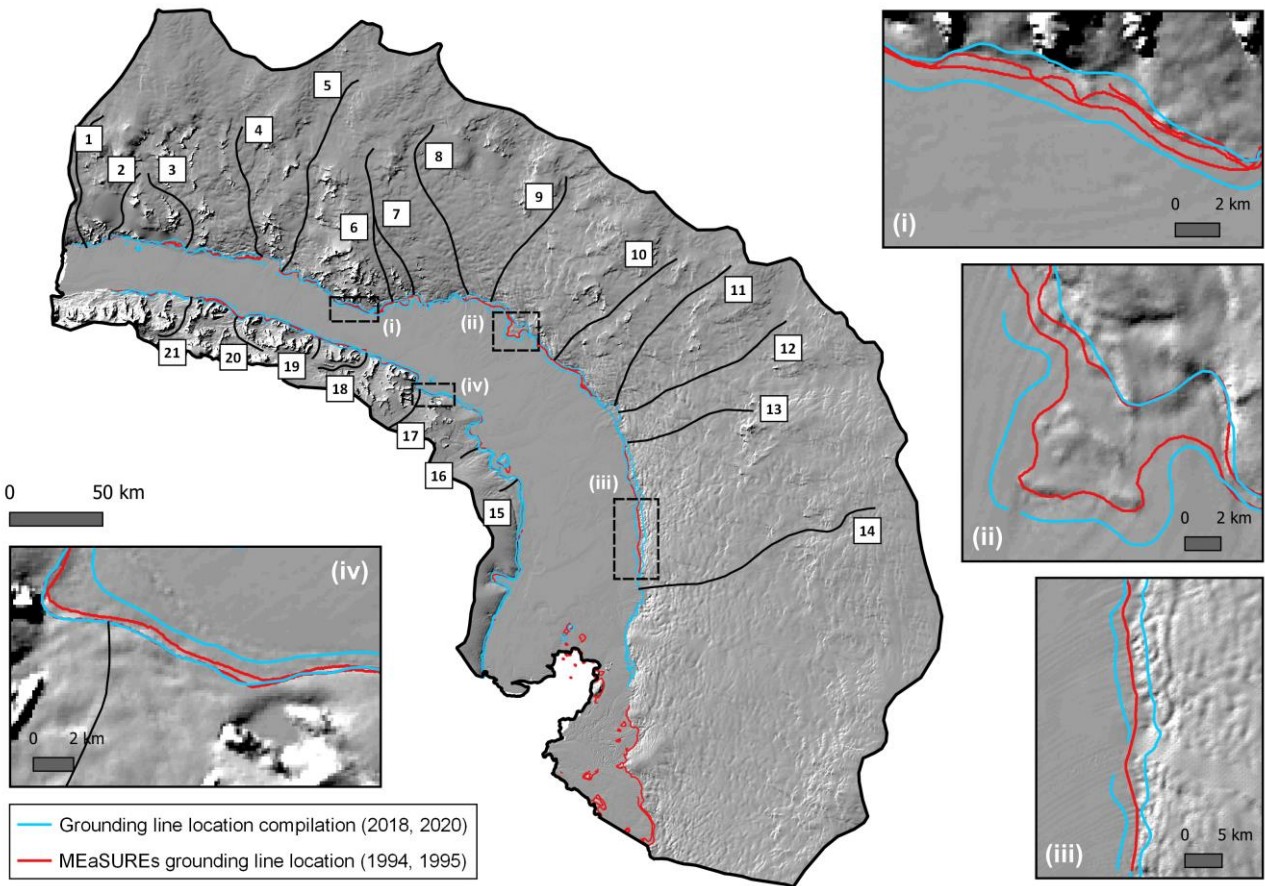

**Figure B1 - Landward (high-tide) and seaward (low-tide) limits of the modern-day (2018-2020) grounding line (blue) and the position of the grounding line in the mid-1990s (red). Background hillshade is derived from the Reference Elevation Model of Antarctica (Howat et al., 2019). Dashed boxes denote the location of the inset boxes (*i-iv*).**

**Appendix C**

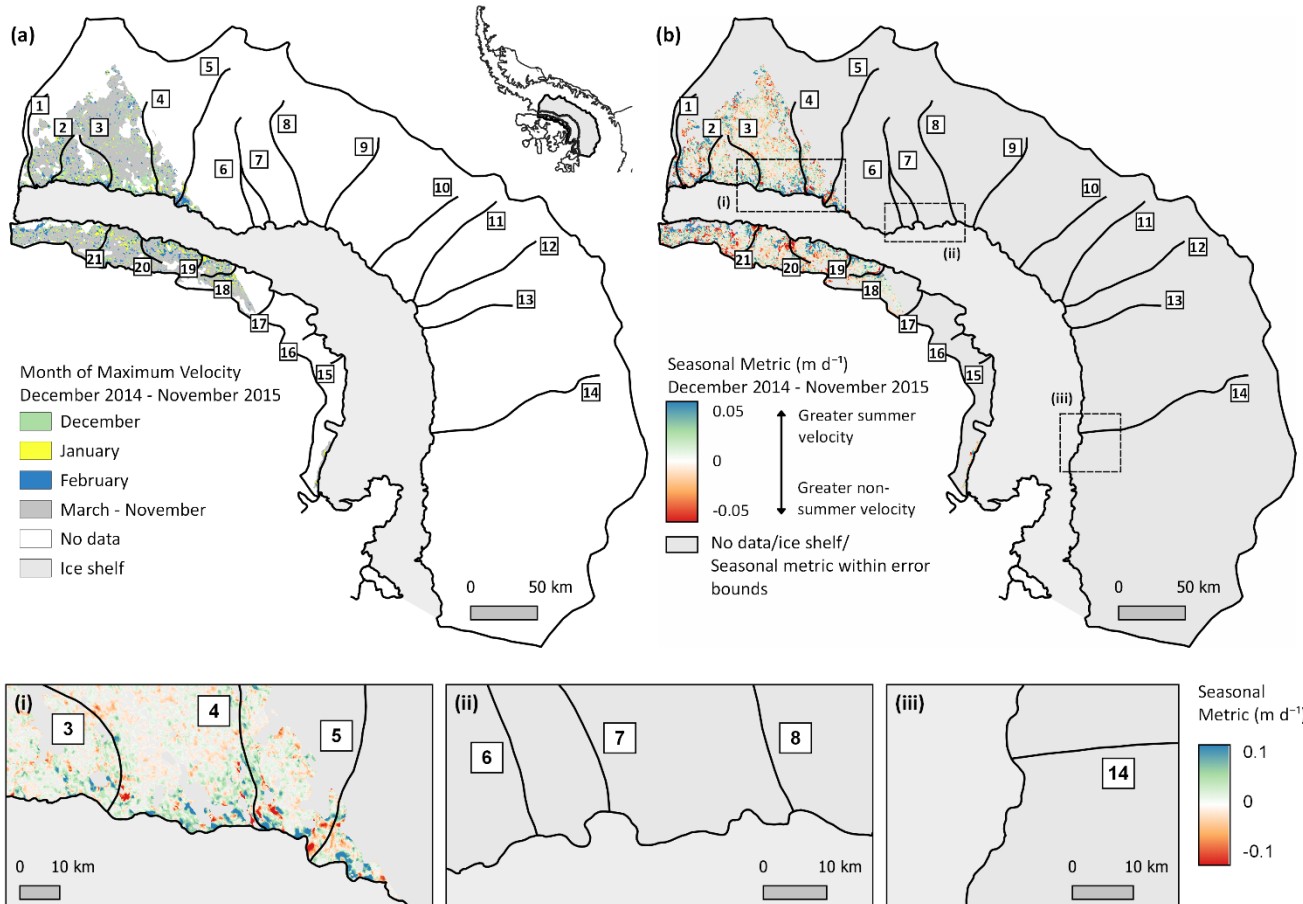

**Figure C1 - Same as Fig. 3 but showing spatially resolved patterns of seasonal ice-flow as observed between December 2014 and November 2015. (a) shows the month of maximum velocity ($v_{time}$); (b) the calculated velocity anomaly ($v_{anom_{SAR}}$). In (b), positive velocity anomalies (blue) indicate summertime speed-up; negative (red), greater velocity during non-summertime months. Dashed boxes denote the location of the inset boxes (*i-iii*). Note that insets ii and iii contain no data for the period December 2014-November 2015. See also Figs. C2-C4.**


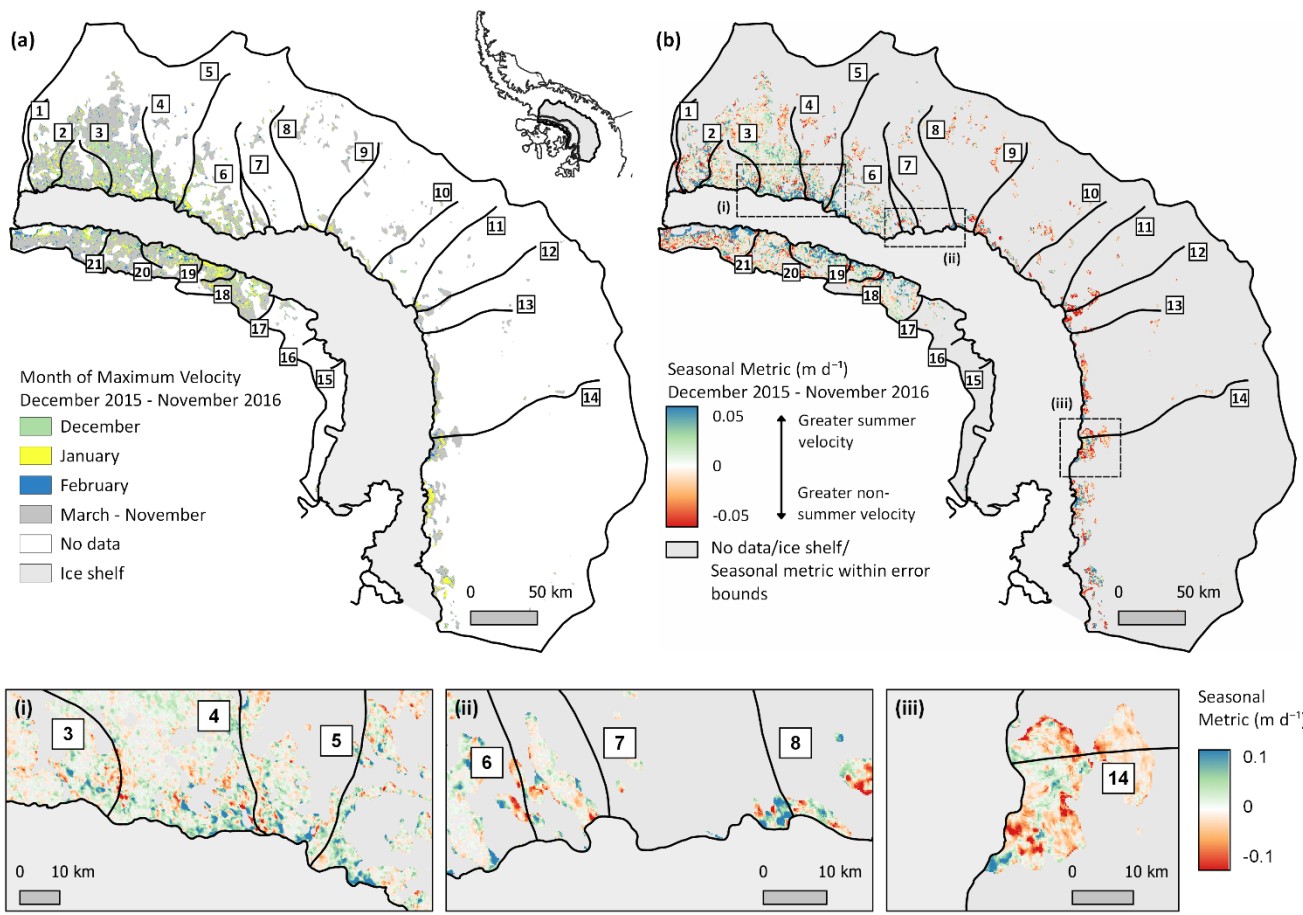


**Figure C2 - Same as Fig. 3 but showing spatially resolved patterns of seasonal ice-flow as observed between December 2015 and November 2016. (a) shows the month of maximum velocity ($v_{time}$); (b) the calculated velocity anomaly ($v_{anom_{SAR}}$). In (b), positive velocity anomalies (blue) indicate summertime speed-up; negative (red), greater velocity during non-summertime months. Dashed boxes denote the location of the inset boxes (*i-iii*). See also Figs. C1, C3-C4.**


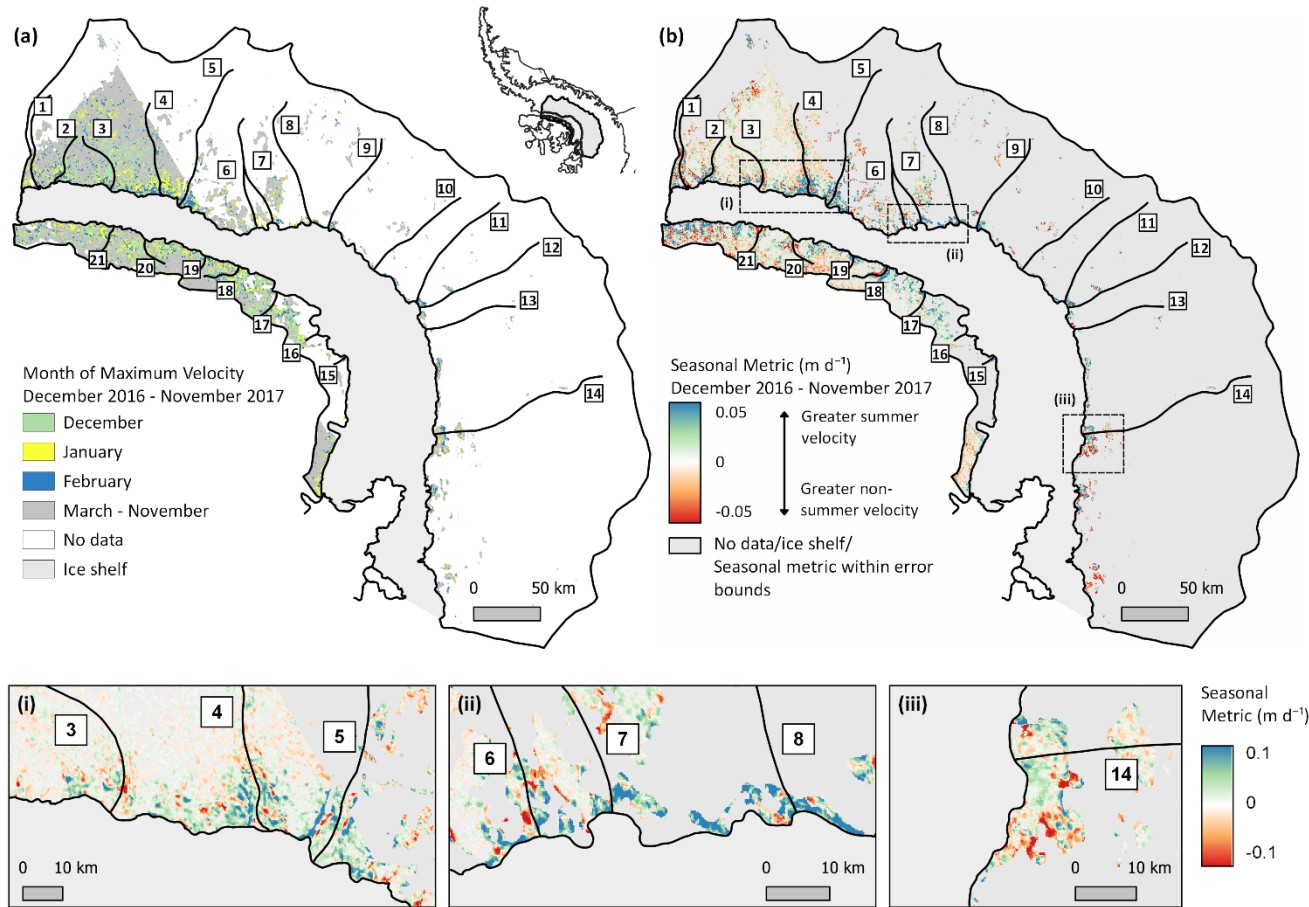

**Figure C3 - Same as Fig. 3 but showing spatially resolved patterns of seasonal ice-flow as observed between December 2016 and November 2017. (a) shows the month of maximum velocity ($v_{time}$); (b) the calculated velocity anomaly ($v_{anom_{SAR}}$). In (b), positive velocity anomalies (blue) indicate summertime speed-up; negative (red), greater velocity during non-summertime months. Dashed boxes denote the location of the inset boxes (*i-iii*). See also Figs. C1-C2 and C4.**


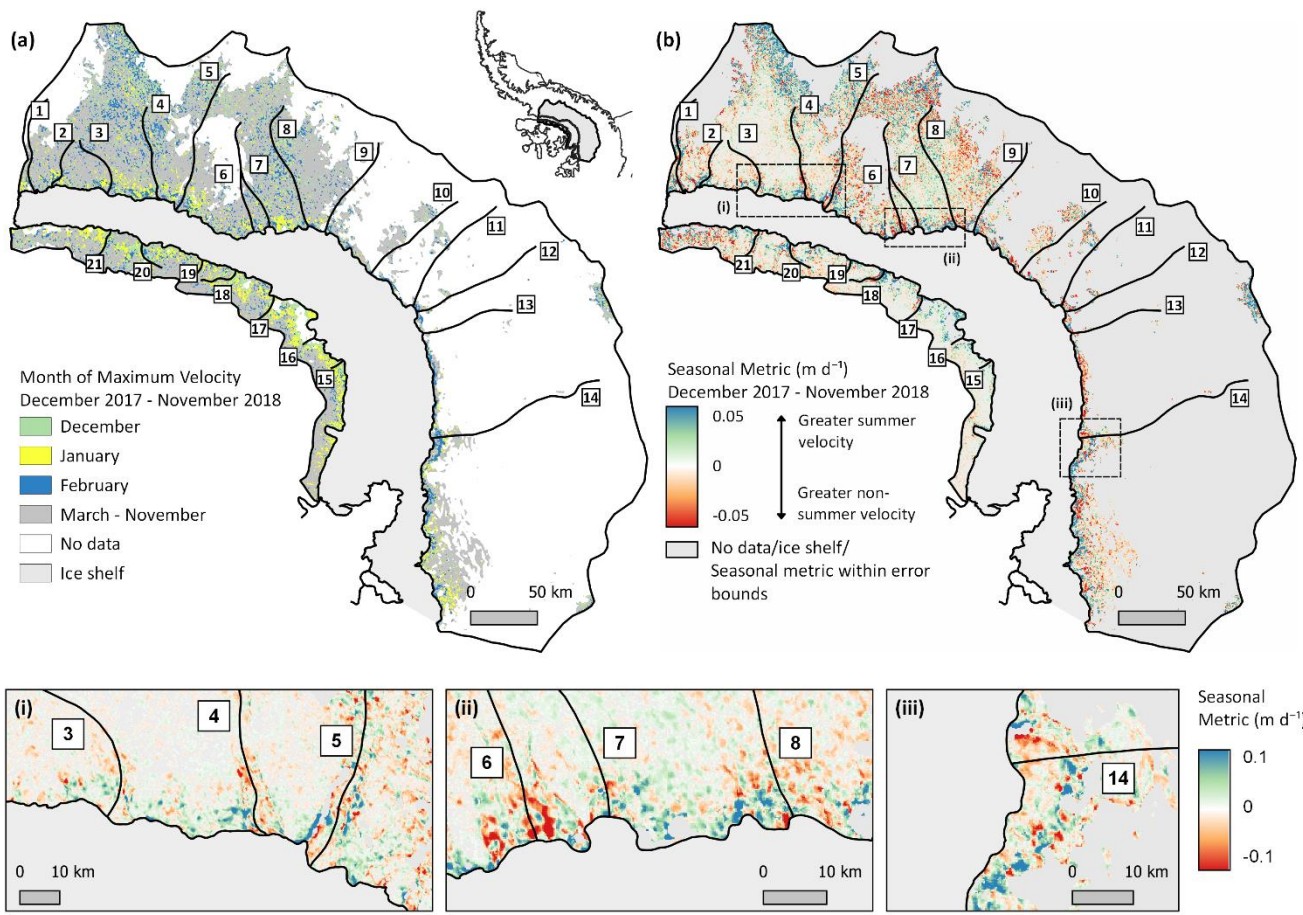

**Figure C4 - Same as Fig. 3 but showing spatially resolved patterns of seasonal ice-flow as observed between December 2017 and November 2018. (a) shows the month of maximum velocity ($v_{time}$); (b) the calculated velocity anomaly ($v_{anom_{SAR}}$). In (b), positive velocity anomalies (blue) indicate summertime speed-up; negative (red), greater velocity during non-summertime months. Dashed boxes denote the location of the inset boxes (*i-iii*). See also Figs. C1-C3.**


**Appendix D**

For most of the outlet glaciers numbered in Figs. 4, 5, A1 and D1, the modelled timing of peak velocity (Table D1) does not align precisely with the month of maximum velocity established from observations (Figs. 4, 5 and 6). To model peak velocity timing, a cosine wave, optimised to the most dominant frequency observed across all outlet glacier time series (1 month), is
fitted to each time series (Fig. D1, left hand panels). The modelled results are subsequently calculated from the phase shift of each fitted cosine wave (Table D1); as such, the modelled timing of peak velocity should be considered a broad-brush indicator of the approximate timing only. The most dominant frequency present in the original function, however, is derived directly from the discrete Fourier transform (Fig. D1, right hand panels), and hence presents meaningful information on the seasonality of each timeseries.


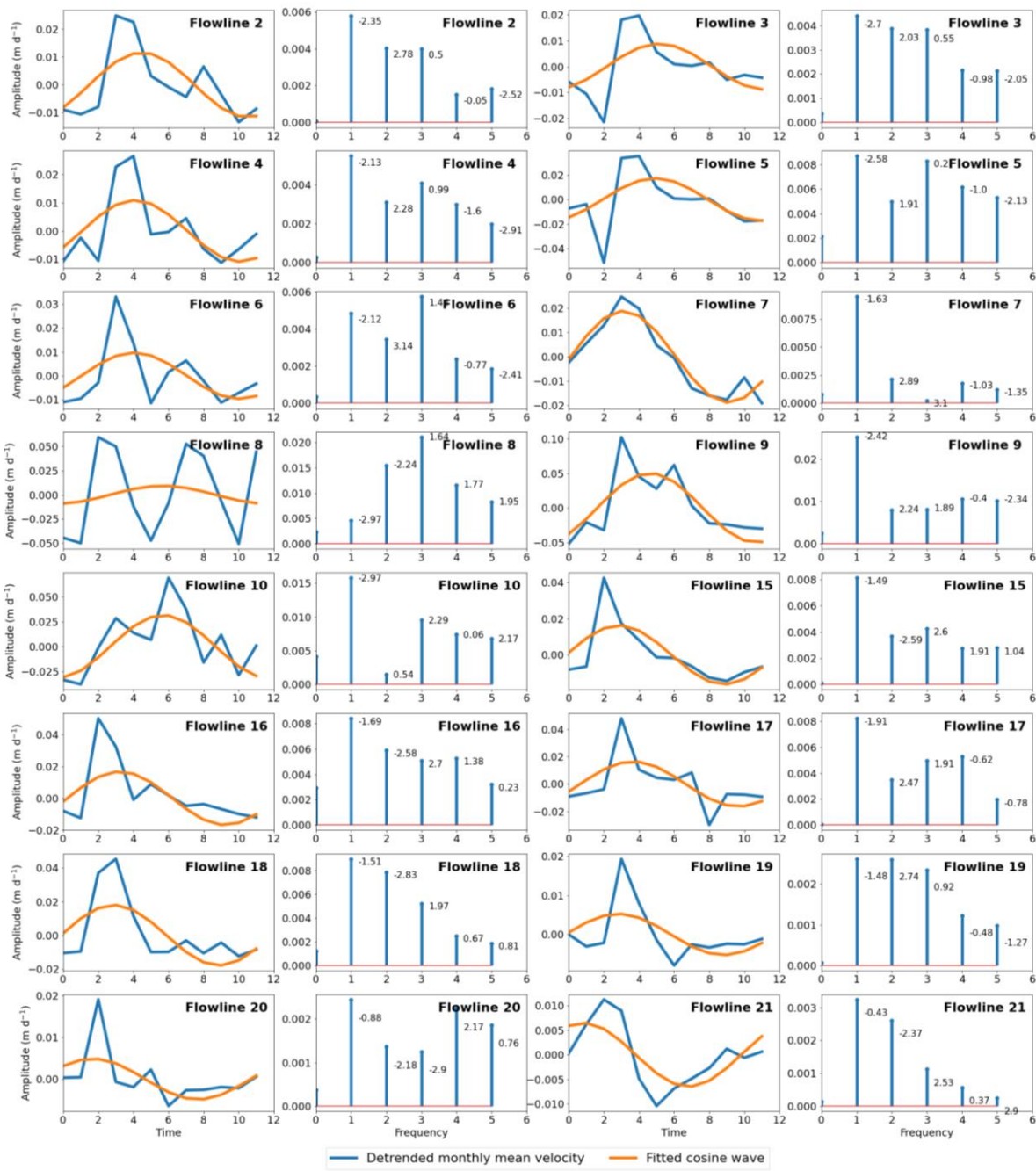

**Figure D1 - Discrete Fourier transform outputs. Left hand panels indicate the time domain where time 0 denotes 1st October. Panels show mean monthly ice flow of each outlet glacier (blue; same as Fig. 5) with fitted cosine functions (orange) optimised to the most dominant frequency observed across all outlet glacier time series. Right hand panels indicate the frequency domain, whereby the greatest amplitude associated with each glacier timeseries denotes the most dominant frequency. In the context of the present study, a dominant frequency of 1 denotes one complete seasonal cycle per year. Annotated values indicate the phase associated with each frequency component.**

**Table D1 - Phase and amplitude characteristics of the fitted cosine function optimised to the dominant frequency determined by discrete Fourier transform analysis (Fig. D1, orange). Flowlines 2-10 drain Palmer Land, while flowlines 15-21 drain Alexander Island.**

| Flowline | Amplitude (A) (m d$^{-1}$) | Phase ($\varphi$) (rad) | Max Occurs (1 Oct + $t$ (months)) | Modelled Timing of Peak Velocity[1] | Season of Peak Velocity[1] | Dominant Frequency |
|---|---|---|---|---|---|---|
| 2 | 0.012 | -2.35 | 1 Oct + 4.48 | 15 February | Late Summer | 1 |
| 3 | 0.009 | -2.70 | 1 Oct + 5.16 | 5 March | Late Extended Summer | 1 |
| 4 | 0.011 | -2.13 | 1 Oct + 4.07 | 3 February | Late Summer | 1 |
| 5 | 0.017 | -2.58 | 1 Oct + 4.93 | 28 February | Late Summer | 1 |
| 6 | 0.010 | -2.12 | 1 Oct + 4.04 | 2 February | Late Summer | 3 |
| 7 | 0.019 | -1.63 | 1 Oct + 3.10 | 4 January | Midsummer | 1 |
| 8 | 0.009 | -2.97 | 1 Oct + 5.67 | 21 March | Late Extended Summer | 3 |
| 9 | 0.050 | -2.42 | 1 Oct + 4.63 | 19 February | Late Summer | 1 |
| 10 | 0.032 | -2.97 | 1 Oct + 5.68 | 21 March | Late Extended Summer | 1 |
| 15 | 0.016 | -1.49 | 1 Oct + 2.85 | 26 December | Early Summer | 1 |
| 16 | 0.017 | -1.69 | 1 Oct + 3.23 | 7 January | Midsummer | 1 |
| 17 | 0.016 | -1.91 | 1 Oct + 3.66 | 20 January | Midsummer | 1 |
| 18 | 0.018 | -1.51 | 1 Oct + 2.88 | 27 December | Early Summer | 1 |
| 19 | 0.005 | -1.48 | 1 Oct + 2.82 | 25 December | Early Summer | 1 |
| 20 | 0.005 | -0.88 | 1 Oct + 1.68 | 21 November | Early Extended Summer | 1 |
| 21 | 0.006 | -0.43 | 1 Oct + 0.82 | 25 October | Early Extended Summer | 1 |

[1]Note: The modelled timing of peak velocity is derived from the phase value associated with each of the numbered outlet glaciers.

## Data Availability

All grounding line and velocity datasets presented in this study are available at https://doi.org/10.17863/CAM.82248 and
https://doi.org/10.17863/CAM.82252, respectively (Boxall et al., 2022a; 2022b). Copernicus Sentinel-1A/B data used in this
study are available from the European Space Agency at https://scihub.copernicus.eu/; Landsat 8 ITS_LIVE velocity data
(Gardner et al., 2019) are available from https://its-live.jpl.nasa.gov/; annual MEaSUREs Antarctic ice velocity maps (Rignot
et al., 2017) are available from the National Snow and Ice Data Center (NSIDC) at https://nsidc.org/data/NSIDC-
0720/versions/1; REMA DEM (Howat et al., 2019) is publicly available at https://www.pgc.umn.edu/data/rema/, and the 2018
grounding line position (Mohajerani et al., 2021) is available at https://doi.org/10.7280/D1VD6G.

## Author Contributions

KB designed the study under the supervision of FDWC and ICW. JW and TN processed the monthly composite SAR-derived
velocity grids. KB performed all analyses. KB wrote the manuscript under the guidance of FDWC, with contributions from all
co-authors.

## Competing Interests

The authors declare that they have no conflict of interest.

## Acknowledgments

This research was undertaken while KB was in receipt of a United Kingdom Natural Environment Research Council PhD
studentship awarded through the University of Cambridge C-CLEAR Doctoral Training Partnership (grant number:
NE/S007164/1). This work was also produced with financial assistance (to FDWC) of the Prince Albert II of Monaco
Foundation, and (to ICW) from the United Kingdom Natural Environment Research Council awarded to the University of
Cambridge (grant number: NE/T006234/1). TN and JW acknowledge support from the European Space Agency through the
Antarctic Ice Sheet Climate Change Initiative (CCI) program. The authors wish to thank W.G. Rees for his advice regarding
the discrete Fourier transform analyses presented in the manuscript.

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
