# Peer review of "Seasonal land ice-flow variability in the Antarctic Peninsula"

_The Cryosphere, 2022_

## Referee Comment (RC2)

**Seasonal land ice-flow variability in the Antarctic Peninsula**

Karla Boxall1, Frazer D. W. Christie1, Ian C. Willis1, Jan Wuite2 and Thomas Nagler2

1Scott Polar Research Institute, University of Cambridge, Cambridge, United Kingdom 2ENVEO IT GmbH, Innsbruck, Austria

5 *Correspondence to*: Karla Boxall (kb621@cam.ac.uk)

**Abstract**

Recent satellite-remote sensing studies have documented the multi-decadal acceleration of the Antarctic Ice Sheet in response to rapid rates of concurrent ice-sheet retreat and thinning. Unlike the Greenland Ice Sheet, where historical, high temporal resolution satellite and in situ observations have revealed distinct changes in land ice flow acregintra-annual timescales, similar seasonal signals have not previously been observed in Antarctica. Here, we use high spattar and temporal resolution Copernicus Sentinel-1A/B synthetic aperture radar observations acquired between 2014 and 2020 to provide the first evidence for seasonal flow variability of the land ice feeding George VI Ice Shelf (GVIIS), Antarctic Peninsula. Our observations reveal a distinct austral summertime (December – February) speedup of ~0.06 ± 0.005 m d-1 (~22 ± 1.8 m yr-1) at, and immediately inland of, the grounding line of the glaciers nourishing the ice shelf, which constitutes a mean acceleration of ~15% relative observation gregional contrasts in the onset of ice-flow acceleration and the overall timing of the speedup events across GVIIS fingerprint oceanic forcing as the primary control of this seasonality. Our findings imply that analogous ice-ocean interactions may be ongoing at the grounding lines of other ocean-vulnerable ounet glaciers around Antarctica. Assessing the degree of seasonal ice-flow variability at such locations is important for quantifying Antarctica's future contribution to global sea-level

20 rise.

**1. Introduction**

Three decades of routine Earth observation have revealed the progressive de revealed in a ccelerated rates of ice thinning, retreat and flow (Gardner et al., 2018; Konrad et al., 2018; The IMBIE Team, 2018; Rignot et al., 2019). This phenomenon has been ascribed to an array of atmospheric and oceanic forcing mechanisms impinging upon

25 the continent (Thoma et al., 2008; Joughin et al., 2012a; Steig et al., 2012; Dutrieux et al., 2014; Paolo et al., 2018), from which resulting land ice losses are estimated to have totalled an average of ~109 ± 59 gigatons per year since 1992 (The IMBIE Team, 2018). Alongside satellite altimetry and gravimetry based assessments of ice-mass change, this trend has partly been constrained from satellite-derived velocity measurements acquired sporadically throughout the year (Rignot et al., 2011a; Mouginot et al., 2012), under the implicit (and unverified) assumption that no discernible intra-annual (i.e., seasonal or shorter)

- 30 variability in ice-flow exists (cf. Greene et al., 2018). In terms of intra-annual ice-flow variability, a historie dearth of systematic, high temporal resolution observations has also limited the ability to search for such changes across Antarctica; this is in contrast to mid-latitude and Arctic ice masses, where the timing and magnitude of seasonal ice-flow variability is now well observed (Iken et al, 1983; Hooke et al, 1989; Zwally et al., 2002; Hoffman and Price, 2014; Moon et al., 2014; King et al., 2018 vitin the context of recent ice-sheet modelling exercises (Seroussi et al., 2020; Edwards et al., 2021), knowledge
- 35 of any intra-annual variations in ice flow is critical for elucidating the processes controlling Antarctica's evolution in a changing climate.

In this study, we examine for and find evidence of seasonal ice-flow variability across the glaciers feeding the climatic vulnerable George VI Ice Shelf, Antarctic Peninsula. We use 6/12-day repeat-pass Copernicus Sentinel-1A/B synthetic aperture radar imagery for this purpose, together with independent, 16-day repeat-pass observations acquired by the Landsat 8 Operational Land Imager. We then discuss the potential mechanisms responsible for driving the observed seasonal ice-flow signals upstream of George VI Ice Shelf, which we ascribe to intra-annual very bility in ocean forcing.

**2. George VI Ice Shelf**

In this study, we investigate seasonal ice-flow variability across 21 glaciers feeding the glaciologically compressive George 45 VI Ice Shelf (GVIIS) (Fig. 1). After Larsen C Ice Shelf, GVIIS is the second largest of the remaining ice shelves fringing the Antarctic Peninsula (Holt et al., 2013), and has an areal extent of 23,500 km2. Ice-shelf flow bifurcates and advects towards both its northern (Marguerite Bay) and southern (Ronne Entrance) ice fronts at an average rate of 0.7 m d-1 (255 m yr-1), with flow averaging 0.08 m d-1 (30 m yr-1) and 1.1 m d-1 (400 m yr-1) along its Alexander Island and Palmer Land margins, respectively (Fig. 1). The thickness of the ice shelf ranges between approximately 100 and 600 m (Morlighem et al., 2020).

---

## Author Comment (AC1)

**Seasonal land ice-flow variability in the Antarctic Peninsula (tc-2022-55)**

**Author Response to Reviewers (13/07/22)**

Dear Dr. Berthier,

We thank both reviewers for their detailed, constructive feedback on our manuscript and were pleased to read that they deem it to be "well structured and ... performed" (Reviewer 1) and one that "breaks new ground on sensitive detection of seasonal velocity signals" (Reviewer 2). We were also especially pleased to read that Reviewer 2, Dr. Ted Scambos, believes the manuscript will provide "a significant contribution to Antarctic glaciology" when published.

Below, we address in full each of the points raised by the reviewers, and have revised the manuscript accordingly. As requested by both reviewers, the most significant change has been to provide a more balanced evaluation of the relative roles of surface and ocean forcing for explaining the observed seasonal speedup signals at GVIIS.

In the following response document, we first present a detailed summary of the major change detailed above (hereafter referred to as **Major Change 1**), and then provide point-by-point responses to the reviewers' detailed comments. We have included the numbered reviewers' comments (*italicised in blue*), our responses (black text) and amendments to the original text (*italicised in grey*). Unless otherwise stated, line and figure numbers referred to below are in line with those in the original manuscript, available at <a href="https://doi.org/10.5194/tc-2022-55">https://doi.org/10.5194/tc-2022-55</a>.

Kind regards,

Karla Boxall

(on behalf of the co-author team)

**Seasonal land ice-flow variability in the Antarctic Peninsula (tc-2022-55)**

**Author Response to both reviewers**

**Major Change 1 – Balanced evaluation of surface and ocean forcing**

Both reviewers raised concerns regarding our attribution of the observed seasonal velocity variation to oceanic forcing, suggesting that the role of surface meltwater should not be dismissed as a potential forcing mechanism. Having reflected on this, we are inclined to agree with the reviewers, with the proviso that there is currently a dearth of surface and ocean in situ records with which to verify the exact mechanism(s) responsible for driving the observed signals. In this regard, we have overhauled the manuscript (Section 5 (Discussion) especially) as suggested by the reviewers to provide a more balanced evaluation (pros and cons) of the relative importance of surface and oceanic forcing.

Specifically, we have made the following revisions:

 a) We have rewritten Lines 16-18 of the abstract to remove the assertion that the ocean is the key driver of the observed seasonality, and instead offer a more balanced statement. The sentences now read:

"Both surface and oceanic forcing mechanisms are outlined as potential controls on this seasonality. Ultimately, our findings imply that similar surface and/or ocean forcing mechanisms may be driving seasonal accelerations at the grounding lines of other vulnerable outlet glaciers around Antarctica."

b) The last clause of Line 42 has been edited to remove the claim that oceanic forcing is the favoured mechanism.

**Sentence now reads:**

"We then evaluate the potential mechanisms responsible for driving the observed seasonal ice-flow signals upstream of George VI Ice Shelf."

c) As outlined above, the Discussion (Section 5; Lines 309-356) has been overhauled to tone down the assertion that oceanic forcing is the sole potential driver of the observed seasonal variability. Similarly, statements proposing that surface meltwater cannot be the trigger of the observed seasonality have been removed. Instead, the Discussion now provides a much more balanced (and nuanced) evaluation of both potential forcing mechanisms.

The revised paragraphs discussing surface forcing (Section 5.1) now note that the timing of our seasonal signal coincides with the peak in annual surface melt rates, and that the greatest speedups (2016/17 and 2019/20) occurred during years in which the melting was particularly intense (Banwell et al., 2021) (on the basis of the comments raised by both reviewers). Reviewer 2 also states that "*It is not essential that the water be visible on the surface as pools*" in relation to previous work that has reported a lack of rapid surface drainage events over the observational era to date (e.g. Rott et al., 2020). We interpret this to refer to the possible presence of firn aquifers not detectable via microwave (and other) remote sensing, a discussion of which – together with the feasibility of such aquifer formation/persistence – is also now included.

Section 5.2 has been similarly revised to tone down the importance of ocean forcing as the sole driving mechanism, and first discusses the existing state of knowledge regarding sub-GVIIS CDW presence and circulation as constrained from temporally limited in situ observations (following the initial version of the manuscript). Addressing reviewer 2's concerns (Comments 31-33), a discussion of the limitations of these in situ records for ascertaining any seasonal cycle in CDW forcing is then presented. This is followed by a discussion of two potential sea-ice driven mechanisms through which modelling studies suggest sea ice variability could alter the depth of CDW presence/influx over seasonal timescales. Finally, a discussion of the cavity throughflow rates needed to drive the seasonal signals is included (in line with Reviewer 2's Comments 3 and 32) as both a 'pro' and 'con' in favour of ocean forcing, motivating the requirement for detailed future in situ data collection within this region of the Southern Ocean for verification purposes.

The revised discussion now reads:

[revised manuscript text omitted]

 d) Finally, we have reworked Lines 361-367 in the Summary and Implications (Section 6) to give a more neutral conclusion regarding our expected explanation of the observed seasonality. This section now reads:

"... Both surface- and oceanic-forcing mechanisms are evaluated as potential controls on this seasonality, although insufficient observational evidence currently exists with which to verify the relative importance of each. Elucidating the precise surface and/or ocean mechanisms governing GVIIS' outlet glacier flow variability is therefore a critical area for future research..."

**Reviewer #1**

1. The authors present a comprehensive analysis of the glacier flow variability of the GVIIS tributaries. The analysis relies on Sentinel-1 data and is backed up with independent Landsat measurements. Overall the paper is well structured and most sections of the analysis are well performed. However, there are some issues that must be addressed:

Most important, the authors state that surface meltwater cannot be the trigger of the observed seasonal variations. However, I am not convinced by the presented justification. Recent publications indicate the warming and also increased surface melt on the AP (e.g. Carrasco et al. 2021, Banwell et al. 2021). So, the authors should also consider surface meltwater in the discussion of their findings or provide evidence that surface meltwater can be neglected as a potential driver. (see also comments below, abstract, discussion, and conclusions need to be adjusted accordingly)

We thank the reviewer for their positive and insightful review, and were pleased to read that they assess the manuscript to be "well structured" and that "most sections of the analysis are well performed".

Having reflected on both reviewers' comments on the manuscript, we agree that at present, surface forcing should not be dismissed as a possible key forcing mechanism (and, likewise, that there is not enough observationally constrained evidence to implicate the primary influence of ocean forcing). With this in mind, we have overhauled the Discussion (Section 5) to offer a more balanced discussion (pros and cons) on the possible surface and ocean forcing mechanisms at work, and have also edited the abstract, introduction (Sections 1 & 2) and Summary and Implications Section (Section 6) to reflect this. These changes are detailed in **Major Change 1** above.

Regarding the papers by Carrasco and Banwell, see also our response to Comment 33 below.

2. Moreover, the description of the methodology has some shortcomings. Please provide here more precise information and be always clear on which region (spatial extent), i.e. whole glacier or just the 10km2 areas, is your analysis and interpretation based. Please justify the interpolation in Fig.6 and explain the applied approach. The error analysis should be also extended. See detailed comments below for some specific issues.

As stipulated in Sections 3.3.1 and 3.3.2 of the manuscript, the analysis presented in Section 4.2.1 is implemented on the entire drainage basin of GVIIS, whereas Section 4.2.2 is executed using the 10 km2 regions immediately inland of the grounding line at each flowline. We have updated the text in Section 3.3.1 to provide further clarification of this point.

**Line 152 of Section 3.3.1 now reads:**

"Following Fig. 2, for each year spanning 2014 – 2019, we first constrained the month in which observed velocities were greatest on a pixelwise basis throughout the entire domain,  $v_{time}$ ."

In Section 3.3.2, Line 170 of the original text states that "we calculated mean monthly velocity and standard error within a 10 km2 region located directly upstream of the grounding line between 2014 and 2020". As such, we believe no further clarification is required in this section.

Please see our responses to the detailed comments below regarding Fig. 6 (Comments 30-32) and the error analysis (Comment 27).

3. Here are also some questions that came to my mind regarding your analysis. Could you please address them? Why is the ice flow higher in March-November for wide regions further inland of the grounding line (GL) and why is it lower during summer?

In Fig. 3a, the colour scheme shown was adopted to emphasise the spatial clustering of summertime (DJF) speedup immediately inland of the grounding line. The raw data used to produce this figure (Boxall et al., 2022b), however, show that a spatially coherent pattern in the timing of speedup does not exist inland of the grounding line, and that the month of maximum velocity is distributed randomly across the twelve months. This random distribution is also reflected in the dataset used to produce Fig. 3b. Therefore, the dominance of seemingly large swathes of grey should not be interpreted as a coherent and/or significant speedup between March and November.

We realise, however, that this was not explicitly stated anywhere in the original manuscript, so we have revised Line 244 to clarify this point. Sentence now reads:

"Notably, no coherent seasonal signal exists beyond ~10-20 km of the grounding line, where  $v_{time}$  is randomly distributed across all 12 months and  $v_{anom_{SAR}}$  shows no associated clustering."

**4. Why is the speedup only visible close to the GL. Why is there [no] speed up further up?**

In our opinion, this is an interesting finding which implicates the role of oceanic forcing, since basal melting at the grounding line would elicit acceleration in flow at the grounding line prior to farther inland. Qualitatively, this is similar to the multi-decadal acceleration trends witnessed both here (see, for example, Hogg et al. (2017)) and in places such as Pine Island and Thwaites, albeit on a much smaller (i.e seasonal) temporal scale. On the assumption that surface melt will be greatest at lower altitudes due to lapse rate effects, the grounding line is where one might equally expect the manifestation of surface drainage-related acceleration to first occur, similar to the seasonal signatures witnessed in e.g. Greenland and Svalbard. However, this mechanism is presumably unlikely to drive such coherent, clustered acceleration events along the *entirety* of the grounding line, as the glaciers which reside there exhibit, for example, highly disparate thicknesses which would require different amounts of surface melting and drainage to enhance basal sliding. These points are now included in our revised discussion (see our comments on **Major Change 1** above).

5. Is there any correlation of speedup with altitude (either the area affected by the speed up or the general hypsometric profile or hypsometric index of the glaciers)? Difference Alexander Island vs. AP?

We thank the reviewer for this interesting question. Fig. 3 shows that the coherent speedup signals we report are restricted to the region at and immediately inland of the grounding line. There, REMA DEM (Howat et al., 2019) also reveals no discernible differences in elevation profile between Alexander Island and Palmer Land. In light of this finding (and that of our related response to **Major Change 1** and Comment 4 above), no further changes have been made to the text.

6. Tides are also affected by the season. Could the seasonal changes of the tides affect the glacier, in particular the GL? E.g. stronger tides lead to a wider grounding zone.

To locate the grounding line in this study, we used the most landward limit of tidally induced vertical ice-shelf flexure observed (see discussion in Sections 3.1 and 4.1). All analyses were carried out inland of this grounding line and thus include grounded ice only, not freely floating ice or the grounding zone.

**Detailed comments:**

7. *l10:* What about the short-term summer speed ups reported by Seehaus et al. 2015 and Seehaus et al. 2016 at Dinsmoor-Bombardier-Edgewoth Glaciers at Sjögren Inlet.

While Seehaus et al. (2015; 2016) focus primarily on the long-term (1993-2014) velocity speedup of grounded outlet glaciers in response to the collapse of Larsen A and Prince Gustav ice shelves, respectively, both studies do also note short-term summertime accelerations of the outlet glaciers. They attribute this phenomenon to the fragmentation of ice mélange, and the associated reduction in buttressing. We have therefore changed the text as follows to account for the findings of these studies.

Line 10 now reads:

"..., observations of similar seasonal signals are limited in Antarctica."

8. L39: Why is it vulnerable?

We have removed the phrase 'climatically vulnerable' from the text.

9. L53: Does the velocity field represent the long-term average?

The velocity field displayed in Fig. 1 represents the mean velocity between September 2019 and August 2020. The caption has been updated to reflect this.

The caption of Fig. 1 now reads:

"Mean ice flow of George VI Ice Shelf..."

10. L56: Source of flowlines?

The flowlines follow the centreline of fastest flow according to the mean ice velocity displayed in Fig. 1. The caption of Fig. 1 has been updated to include this information.

**The following text has been added to the caption of Fig. 1:**

"Numbered flowlines delimit the centreline of the fastest-flowing outlet glaciers draining to GVIIS (named according to the UK Antarctic Place-names Committee) ..."

11. L69: You list publications regarding meltwater lakes from 2017 onwards and say that such studies lead to the identification of GVIIS as a potential site for future ice shelf disintegration, identified in a study from 2013. That's somehow inconsistent

Thank you for this comment. For chronological consistency, the reference citing GVIIS as a potential site for future disintegration has been updated from Holt et al. (2013) to Holt and Glasser (2022).

**12. L85: please explain "seaward extent". The glaciers are flowing into an ice shelf.**

The Southern Ocean extends beneath GVIIS. We would therefore prefer to keep this term in the text.

**13. L98: Did you apply any multi-looking or filtering? What about the coregistration of the images? Some more technical information would be nice.**

We have added additional information on the co-registration and geocoding of the SLC images and interferograms (cf. Comment 17 below) to this section, as well as the inclusion of a reference to Christie et al. (2022) which provides further technical details. We also note that further information is provided in the metadata/README files associated with the grounding line datasets accompanying

this paper (Boxall et al., 2022a). Beyond the use of REMA DEM at the co-registration, DInSAR and geocoding steps in this study, all other processing parameters are virtually identical to those cited intext, so for brevity we would prefer not to re-list these here.

**Revised text reads:**

"To recover the location of GVIIS' modern-day grounding line, we employed double-differential interferometric SAR (DInSAR) processing techniques to all consecutive 6-day repeat-pass Sentinel-1A/B Interferometric Wide (IW) single look complex (SLC) images acquired during extended austral wintertime (May-October) 2020. Extended austral wintertime imagery was used to maximise phase coherence between successive image pairs which may be degraded due to the attenuation of radar waves by summertime supraglacial water presence (cf. Sect. 2). Similar to earlier work (Park et al., 2013; Rignot et al., 2014; Christie et al., 2016; 2022), we co-registered each successive image pair and removed the topographical component of phase from all subsequently generated interferograms, using the Reference Elevation Model of Antarctica (REMA; Howat et al., 2019). Assuming ice creep to be common between each SAR image, we then differenced all successive interferograms to locate the limit of tidally induced vertical ice-shelf flexure. This limit is represented as the landward extent of a band of closely spaced fringes on double-differenced interferograms (Rignot et al., 2011b), and is an accurate proxy for the true grounding line which cannot be recovered directly by satellite-based imaging techniques (Fricker et al., 2009; Friedl et al., 2020). Finally, all interferograms were geocoded to EPSG:3031 (Antarctic Polar Stereographic projection) using REMA."

**14. L100: Could you please provide an overview of the used imagery**

Every available Sentinel 1A/B IW SLC image available between 1st May 2020 and 31st October 2020 was processed to delineate the grounding line location. This is stated in Lines 98-99 in the text, and so no further changes have been made to the text.

**15. L103: Did you prove this assumption? You should use your velocity measurements to prove it.**

This assumption is part of a standard technique reported commonly in the literature. We would therefore prefer to keep this sentence as originally found in the text. We appreciate, however, that velocities will not be entirely common over these periods (Rack et al., 2017) and that this can induce grounding line errors of up to one ice thickness. These errors, however, fall inside the total range of grounding line positions imaged (Fig. B1; Boxall et al., 2022a).

**16. L119: Please describe here briefly how the uncertainty was estimated and what is a "valid pixel". This would be beneficial for the reader**

As indicated in the text (Line 119), uncertainty was estimated as one standard deviation  $(1\sigma)$  from the mean of all valid pixels. We therefore believe no further action is required on this point.

**Regarding valid pixel count, we have now clarified this on Line 119. Revised text reads:**

"... associated grids of uncertainty (1 $\sigma$ ) and valid pixel count (the number of non-NaN observations used in the production of each monthly estimate)."

**17. L122: This information should be provided in section 3.1. and here you can refer to 3.1.**

**Done. See also our response to Comment 13 above which includes details of the revised text.**

Reference to Sect. 3.1 has been added to Line 123.

**18. L127: Here you can refer to Friedl et al. 2021 as well. Their study is based on the same satellite data.**

Thank you for this suggestion. This reference has been added to the text.

**19. L130: what is sigma? The average of all pixels?**

This is a similar question as that posed in Comment 16 above. Sigma ( $\sigma$ ) is the standard notation used to denote standard deviation. This is stated explicitly in Line 131.

20. L136ff: Unclear explanation. You are using intensity tracking, thus you measure also displacements in azimuth direction and not only in range (LOS) direction. For sure, the shifts in the phase center depth can affect your measurements. But please rephrase this section to be more clear. Did you account for this shift in LOS direction? How much would it be? Any suggestion on how to estimate the bias? A brief statement would be nice at the end of this section.

We thank the reviewer for highlighting this small typo (LOS pertains to the slant range of the sensor, and not the two (range/azimuth) components required to estimate horizontal displacement) which the second reviewer (Ted Scambos) also remarked upon (see his Comment 24 below). For clarity, we have reworked this section of the text to read:

"We averaged velocities over monthly timescales to minimise contamination associated with ionospheric and tropospheric delay between successive (6/12-day repeat pass) Sentinel-1 image acquisitions (Rosen et al., 2000; Selley et al., 2021). Moreover, across the Antarctic Peninsula, Sentinel-1A/B acquisitions are currently acquired in descending mode only (ESA, 2022), meaning that the velocity data utilised in this study represent the relative displacement of point targets from a single look angle only. Recent work has shown that below sub-monthly resolution, these velocities can be subject to bias owing to radar penetration differences associated with the freezing state of the snowfirn-ice interface between image acquisitions (Rott et al., 2020). This phenomenon can induce shifts in the radar line-of-sight (LOS) distance to target by up to several meters (Joughin et al., 2018; Rott et al., 2020), resulting in either an under- or overestimation of velocities depending on the flow direction of the ice relative to LOS (Rott et al., 2020). At present, reliable, sub-monthly estimates of the magnitude of this bias over the western Antarctic Peninsula are difficult to model owing to a lack of detailed insitu information on the temporally variable composition of the firn layer; for these reasons, monthly composites were also utilised to dampen the effect of this bias."

Regarding firn-related phase shifts, we believe our initial writing made it implicit that the use of monthly composites was intended to minimise sub-monthly bias(es), although for clarity we have now provided more detail on the difficulties associated with trying to model/correct for this potential phenomenon in the last sentence above.

**21. L152: On which spatial scales did you apply the analysis. Throughout the whole glacier area? Only for the 10km2 areas next to the GL? Please clarify**

This comment is addressed in our response to Comment 2 above.

**22. L153: Is this analysis based on the monthly mosaics or single velocity fields?**

As stated on Line 118, all analyses were carried out using the monthly mosaics. Monthly mosaics were used instead of the single velocity fields to reduce atmospheric contamination (Lines 133-134) and minimise the effect of the bias caused by the shift in the LOS distance (added to Line 141 in response to Comment 20 above). As such, we believe no further action is required on this point.

**23. L165: Do you remove pixels that had no coverage for a specific month or even for single SAR image pairs? Please clarify.**

Pixels with no coverage for a specific month (i.e. a valid pixel count of 0) were removed. The text has been updated to clarify this.

**Line 165 now reads:**

"... pixels without continuous monthly data coverage throughout the entire year were culled from our analyses."

**24. L167: What about very slow-flowing regions? Will they be discarded? (or did you analyze fast-flowing regions only?, see comment above)**

We believe this comment pertains to Section 3.3.2 as a whole. In short, yes, we have discarded very slow-flowing regions. We have reworded this section's header to clarify this point, noting that an examination of slow-flowing regions would be illogical in the case of the present study given the lack of any coherent seasonal signal in the slower flowing regions. New heading reads:

"3.3.2 Flow variability near the grounding line of GVIIS' fast-flowing outlet glaciers"

**25. L193: feature tracking**

Thank you. The typo has been corrected.

**26. Fig.3: Why is the pattern so noisy? Any explanation? Could you also include the glacier numbers in the upper maps?**

The spatial pattern displayed in Fig. 3 is noisy because it represents a complex series of geophysical processes. Per SAR image acquisition, many such processes contribute to speckle returns on a per pixel basis, including (but not limited to): firn effects, snow blow and surface temperature, height slope change and, of course, ice dynamical processes. Changes in speckle typically reduce coherence between successive scenes, which will ultimately propagate into our monthly averaged velocity estimates. While we could have performed filtering and/or smoothing of these grids to reduce the noise in the figures, we were eager to avoid manipulating the "raw" observations as much as possible so as to circumvent the presentation of potentially biased and/or unrepresentative seasonal signals.

The outlet glaciers have been numbered in Fig. 3, as suggested.

27. Fig.4: How did you compute the error bars? How did you compute the mean monthly velocity for the period 2014-2020? Please provide more information or a link to the respective section. Fig. A2 indicates that for several glaciers the availability of monthly means was quite limited (1-3 measurements, e.g. flowline 16, 10 ...) How did you account for this issue in your analysis?

To address the first element of this comment, we have revised Line 170 in the text. The sentence now refers to Sect. 3.2 which explains how the error was calculated (see Equation 1), and clarifies how the mean monthly velocity was calculated.

**Line 170 now reads:**

"To do this, we calculated the mean velocity and standard error (cf. Sect. 3.2) at monthly intervals within each 10 km2 region located directly upstream of the grounding line between 2014 and 2020."

Regarding the second element of this comment, standard error is reported specifically for the reason identified in the reviewer's comment, since it implicitly accounts for variability in data availability as

determined from the valid pixel count (Sect. 3.2, Equation 1). This is a standard method in which to report error associated with velocity measurements (see, for example, Rignot et al. (2016) and Greene et al. (2018)), so we believe no further justification of the technique is required in the text. On a related note, despite the lack of observations at some glaciers, there are still obvious summertime speedup signals at those locations which are completely consistent with the timing and magnitude of behaviour exhibited elsewhere. This, by implication, verifies the behaviour of the more poorly observed glaciers (and vice-versa).

28. L262: Maybe there was a switch between effective and ineffective subglacial drainage. This might explain the late-summer slowdown. At some other glaciers, a late summer or even March/April minima is also visible.

And the late winter slowdown might be caused by a lack of bed lubrication at all. Well, that is just pure speculation from my side. Some studies at Columbia Glacier or also in Greenland revealed similar patterns. (e.g. Moon et al 2014, Vijay and Braun 2017...)

The authors are aware of the meltwater-controlled seasonal velocity variations observed on the Greenland Ice Sheet (e.g. Moon et al., 2014) and mountain glaciers (e.g. Vijay and Braun, 2017) that are attributed to the switching between inefficient and efficient subglacial drainage networks. Having taken on board the comments made by both reviewers regarding the possible importance of such surface forcing mechanisms, we have now discussed the pros and cons of (amongst others) evolving drainage efficiency in the Discussion section. Please see our responses to **Major Change 1** for further details.

29. \*L271: Maybe surface melt onset is earlier on Alexander Island as compared to the glacier's origination from the AP. Any correlation with average glacier altitude or surface melt data from climate modeling data?

This is a good question, although as stated in our revised Section 5.1 (see **Major Change 1** for further details), passive microwave records show no obvious differences in total summertime melt rate between Alexander Island and Palmer Land (cf. Trusel et al., 2013; Bell et al., 2018; Johnson et al., 2022). While this of course may not *always* be true owing to, for example, very short-lived passing storms/atmospheric rivers, it is consistent with the Rossby radius of atmospheric deformation at that latitude (~1000-1500 km wide). In other words, the climate and any associated melting would, over seasonal (and longer) timescales, be expected to be largely homogeneous at Palmer Land and Alexander Island inland of the GL, which are <70 km from one another.

The suggestion that there may be a correlation with altitude has been addressed our response to Comment 5.

**30. L287: Why did you apply any interpolation? Just show the pure data.**

We thank the reviewer for allowing us the opportunity to revise the language used here. We realise that 'interpolation' is probably the incorrect choice of word in this instance, given that mathematical interpolation was not actually applied in the production of Fig. 6 (now Fig. 7). Instead, Figure 6 (now Fig. 7) merely represents a gridded statistical summary of the density of valid pixels, and does not manipulate the underlying data in any way. Ultimately, this visualisation was chosen to better highlight high density regions given the noisy nature of the original data (see also our response to Comment 26).

The text has therefore been reworded to better articulate this fact. Lines 286-291 now read:

"Upon examination of the velocity field (Fig. 1) in conjunction with the phenomena observed in Fig. 7, we infer the clustered regions of agreement farther inland (~40-150+ km from the grounding line) to be falsely identified regions of speedup falling close to combined sensor error limits (cf. Sect. 3.3.3). There, clustering resides mostly over areas of near-stagnant flow unlikely to have experienced significant seasonal variability (~10 m yr-1)."

**Note that Figure 6 is now Figure 7 following the insertion of Figure A1 to the main text (now Figure 4).**

31. Fig.6: Please use different colors or line styles to illustrate the SAR derived average velocity contours. How did you generate the heat-map? Please provide more information on how you computed the density. Please do not interpolate the density, if the interpolation is causing such strong artifacts (see comment above).

We have revised Fig. 6 (now Fig. 7) to use different line styles for the SAR-derived velocity contours.

As described in our response to Comment 30, the densities displayed in Fig. 6 (now Fig. 7) are not interpolated, but reflect the densities of unmanipulated observations within a prescribed search radius (10 km). Since we mask all observations seaward of the grounding line, the search radius used to calculate density is effectively halved relative to further inland. Mathematically, this explains the circular-like signatures observed inland, which extend beyond the spatial dimensions of fast flow and are therefore believed to be unrealistic (see revised text in our response to Comment 30).

For reasons of data transparency, we would prefer to retain the figure as is (rather than, for example, generating a new figure where these obvious blunders are clipped out). We would, however, be happy to reconsider should the reviewer and/or editor feel strongly in support of this option as an alternative.

**32. L291ff: Please show at least one example in Fig.6. Otherwise, it is difficult to figure out this issue.**

Done. A few symbols have been added to Fig. 6 (now Fig. 7) to highlight examples of the circular-like artefacts mentioned in the (revised) text, and the caption has been adjusted accordingly.

**Caption reads:**

"Red crosses represent examples of areas of clustering not representative of a true geophysical signal."

33. L320: You should also mention the more recent warming on the AP which overlaps strongly with your observation period (reported by Carrasco et al. 2021). This should be considered in your discussion. There is also a strong surface melt anomaly in 2019/2020 reported by Banwell et al. 2021 on Alexander Island and at least close to the GL next to the AP. So you should consider also the option of surface meltwater as a driver for seasonal fluctuations

We thank the reviewer for raising these points, and reiterate that the majority of this comment has now been addressed in **Major Change 1**.

Regarding the recent findings of Carrasco et al. (2021) and Banwell et al. (2021), this is a similar comment to that posed in Comment 1 above. In short, we would prefer not to include Carrasco et al. in the references given the inconclusive evidence presented in that study for the possible return to relatively warmer conditions over the Antarctic Peninsula. The basis of their argument pertains to the one or two anomalous warming years seen in recent years (2016/7 and 2019/20) although, from a climatological perspective, it is simply too early to ascertain whether this is representative of a statistically significant trend.

Regarding the study by Banwell et al. (2021), the anomalous warming the reviewer notes is restricted largely to the Wilkins Ice Shelf region (i.e., the other side of the ice divide from our region of interest); this is consistent with the climatological limit of ice shelf viability as noted in our revised Discussion (cf. Section 5.1 and references therein). Unfortunately, no data exists across most of Palmer Land in that study, so any detailed comparison between 2019/20 island vs. mainland melt would furthermore be scientifically unrobust. We do, however, now make note of the seemingly strong correspondence between the magnitude of our accelerations and the 2019/20 melt event documented by Banwell et al. (2021) in the revised discussion.

**34. L324ff: Please revise and account also for potential surface meltwater availability (see comment above)**

We thank the reviewer for raising this point, which we have addressed in our response to **Major Change 1** above.

35. Fig. A1: On many panels, the glacier names are covered by black lines.

Thank you for this comment. The label positions have been adjusted so that they no longer overlay the data.

36. Fig. B1: Could you please include the central flowlines of the glaciers and glacier numbers.

Thank you for this suggestion. Numbered flowlines have been added to Fig. B1.

The flowlines have also been numbered in Fig. 3 and Appendix C, for consistency.

37. Fig.C...: what about 2019-2020?

The SAR-derived velocity observations extend from October 2014 to August 2020. For the calculation of  $v_{time}$  and  $v_{anom_{SAR}}$ , a year was considered to span from December to November (i.e., four complete seasons). Therefore, the calculation of these metrics for the period encompassing December 2019 – November 2020 was not possible because the processed velocity time series only extends to August 2020.

The following sentence has been added to Line 166 to clarify this:

"Notably, the calculation of either metric was not possible for 2019/20 given the lack of velocity data beyond August 2020 (Sect. 3.2)."

38. Table D1: Could you also include the most dominant frequency

Done.

**Reviewer #2**

1. The paper uses a combination of Sentinel 1 A/B velocity data and ITS\_LIVE Landsat-8 velocity data, along with a careful mapping of the grounding line, to assess the scale and extent of a clear seasonal variation in flow speed for glaciers inflowing on both sides of the George VI ice shelf. The paper makes a strong case for the validity of the signal they see, and the seasonality is quite sharp and clear, albeit not large. The authors attribute this to variation in ocean forcing. This is a well-written, well illustrated and described study that breaks new ground on sensitive detection of seasonal velocity signals (a -few- other studies are out there now for some other regions).

We thank the reviewer, Ted Scambos, for his detailed and constructive review and were particularly pleased to read his endorsement that our study is "*well-written, well illustrated*" and "breaks new ground on sensitive detection of seasonal velocity signals".

2. However, the attribution of this signal to ocean forcing in untenable. While this means that the paper absolutely needs to be revised, in fact 80% of the paper is ready to go. It is necessary that the paper revise the attribution to discuss the pros and cons of ocean forcing and surface melt percolation to the bed equally. That is, if the following considerations do not convince the authors that surface melting has in fact a far stronger case for this speed-up. I would like to point out that such a conclusion, or preferred but qualified causal process (surface meltwater reaching the glacier bed), would still make this paper a significant contribution to Antarctic glaciology.

This is a similar verdict to that expressed by Reviewer 1, which has formed the basis of our **Major Change 1** above. As detailed in that section, we have adhered to both reviewer's suggestions and have overhauled the manuscript (Discussion (Section 5) especially) to provide a balanced discussion of the pros and cons of both surface and ocean forcing. Ultimately, we hope the reviewer will concur that these are underpinned by a lack of in situ observational evidence needed to unambiguously ascertain the leading driver(s) of these signals, which is also now reflected in our revised manuscript.

3. The sharpness and regularity of the signal, spanning the entire GVIIS cavity within one or two months (Figure 4 and 5), is the first indication that this is related to summertime melt rather than ocean flow. Peaks in ice flow in December and January are timed closely with peaks in surface melting. Moreover, these timings occur sharply year after year (Figure A1). Nearly all of the glaciers showed a significant spike in 2019-2020, a major melt year for the region. As the paper notes, the -potential- for surface water to induce glacier acceleration is well-proven. It is not essential that the water be visible on the surface as pools (see papers by Harper, Humphrey, Pfeffer; by Koenig, O. Miller, Miégè, Forster.)

On the other hand, oceanographic signals along the Antarctic coast are rarely so sharply seasonal. The cited papers do not (-can- not) discuss seasonal variations in cavity currents or changes in the depth of the CDW layer. The authors infer and favor ocean forcing, but don't discuss how it would occur – would it be related to sea ice losses? (far more variable and uncertain than the surface melt season) Or wind patterns moving the polar water layer and changing isopycnals in that fashion? (also not reliable enough to provide a signal like Flowlines 2, 3, 4, 5, 17, 18, 19, 20, and 21 in Figure A1). Note that if the change in CDW depth or flux is related to a south-to-north current, the speed required would be an order of magnitude faster than that discussed in Jenkins and Jacobs, 2008 (and it would have to be a continuous laminar flow or wave in the isopycnal).

We thank the reviewer for these comments, most of which are also reflected (and addressed) in the reviewer's subsequent comments below and in our responses to **Major Change 1**.

In short, most of the points raised here regarding surface melt vs. oceanographic forcing have now been incorporated into our revised discussion. We wish to reiterate here, however, that while we appreciate the sentiments expressed regarding ocean forcing, we are less convinced by the reviewers' comments on the lack of sharp seasonality in ocean state (see, for example, the papers on strong ocean seasonality along the Antarctic coast by Holland et al. (2010) and Webber et al. (2017) referenced in the Discussion (Section 5)). Our initial version of the manuscript also included suggestions for the processes hypothesised to be driving these signals, making explicit reference to these papers (Lines 336-356). It further occurs to us that the strong velocity signals observed may indeed be forced by the ocean, but only 'sharp' in nature due to the competing influence of across shelf glaciers (cf. Comment 5 below). That said, following our response to Comment 2, a lack of in situ data exists to verify this speculation (and the relative importance of both surface and ocean forcing processes in general), resulting to our decision to opt for a more balanced (pros/cons) approach in the revised discussion.

4. At the very least the authors need to discuss the two possibilities as equally likely. Personally, the case for summer melt influence is far stronger in my view. However, there are data that might save the ocean discussion: instrumented seals. Data collected by instrumented seals and analyzed by, e.g., Lori Padman or Lars Boheme, might be able to show strong seasonal ocean variations. Have a look at Padman et al., 2012, JGR-Oceans – perhaps in the data used in that paper there is an indication of seasonality (but I don't think it is mentioned in the paper). The sharp downturns in the ice velocity just before, or just after, the seasonal speed-up pulse are not easily explained in the ocean scenario.

We are grateful for these suggestions and, following **Major Change 1** and our responses to the reviewer's Comments 2, 3, 6 and 33, we have now overhauled the discussion as suggested. The use of seal data is also a good suggestion and one which we are already examining for a follow-up piece of research going beyond the 'proof-of-concept', satellite-based remit of this study.

5. Also – the authors missed something really cool in the data shown in Figure A1. Look carefully at the signal of Flowline 3 and Flowline 21, and their geographic position. These glaciers are influencing each other across the ice shelf. The earlier acceleration of Flowline 21 (Grotto Glacier, west side, Alexander Island) -slows- the outflow of Flowline 3; then Flowline 3 (Millet Glacier, east side, with a later melt-season peak, perhaps?) accelerates and forces Flowline 21 to slow down. You can see a similar but less clear influence in Flowline 2 and 4, and then Flowline 20 and 19. Like an angry uncle at Thanksgiving, one glacier is shoving the ice shelf table, turkey and all, towards the unsuspecting nephew; the nephew then makes his final point, and shoves the table back toward the uncle. (I suppose it stars with the aunt dumping her drink, meltwater, on each of their heads in succession.)

We thank the reviewer for this metaphor which, as Europeans, was entertaining. Family feuds aside, this is a highly astute and interesting observation that we have now included in the revised text in Section 4.2.2. Lines read:

"Spatially, our results are also suggestive of an apparent regional contrast in the timing of summertime speedup ... Within this trend, Figs. 4 and 5 also present clear evidence of the ability of GVIIS' outlet glaciers to influence each other across the ice shelf, whereby the earlier acceleration of Alexander Island's glaciers initially arrest those flowing from Palmer Land, delaying the onset of their acceleration until the late summertime (compare, for example, the timing of peak velocity at the geographically opposite Flowlines 21 and 3, and Flowlines 20 and 4-5). Upon late-summertime acceleration, Palmer Land's glaciers then arrest the flow of those on Alexander Island in a similar manner."

On the basis of this observation, we have also now decided to place Fig. A1 into the main text as the new Fig. 4, and have updated all other figure numbers accordingly.

6. The last parts of the paper should be re-written with these considerations in mind, but the majority of the paper is publishable as is. I suggest moving some of the figures to the appendices or supplemental information, but overall this is a very well done study, that needs to revise the attribution to a wider perspective at least; if not outright favor surface melt-driven acceleration.

*I would like to review the revised paper. Also, the authors are invited to Thanksgiving at my house.*

**Thank you for the invite! We very much look forward to joining you in November!**

**Detailed Comments**

7. L10 – Change to ...within intra-annual timescales...

**Thank you for this suggestion. This edit has been implemented.**

**Line 10 now reads:**

"Unlike the Greenland Ice Sheet, where historical, high temporal resolution satellite and in situ observations have revealed distinct changes in land ice flow within intra-annual timescales, ...".

**8. L16 – using what satellite or data set?**

This is a good suggestion. We have named the satellite used to derive the independent optically derived velocity observations.

**Line 16 now reads:**

"These findings are corroborated by independent, optically derived velocity observations obtained from Landsat 8 imagery."

**9. L18 – Can you add a short statement about the basis for this statement? Beyond contrasts in onsets and overall timing – something about the facts that led you to this?**

**These sentences have been revised following **Major Change 1** to provide a more neutral discussion of the relative roles of surface and ocean forcing. New lines read:**

"Both surface and ocean forcing mechanisms are outlined as potential controls on this seasonality. Ultimately, our findings imply that similar surface and/or ocean forcing mechanisms may be driving seasonal accelerations at the grounding lines of other vulnerable outlet glaciers around Antarctica ...".

**10. L22 – demise is too strong a word... for now. Decline?**

Thank you for this comment. We have replaced the word demise.

Line 22 now reads:

*"Three decades of routine Earth observation have revealed the progressive decay of the Antarctic Ice Sheet, ..."*

11. L25 – add something from surface melting and fracturing here – Rignot et al., 2004; Cook and Vaughan

**Thank you. Both studies have been added to the cited literature here.**

12. L30 – a general dearth...

Thanks. We have replaced 'historic' with 'overall'.

13. L34 – need to note that the signals are much larger for these more northern ice masses – or we'd have seen them already

**Line 32 has been edited to highlight the large magnitude of the seasonal signals observed on midlatitude and Arctic ice masses.**

"...; this is in contrast to mid-latitude and Arctic ice masses, where the timing and large magnitude of seasonal ice-flow variability is now well observed."

**14. L37 – change to: In [this] study, we find evidence of...**

Done.

**15. L37 – remove climatically – you're claiming that it's being impacted mostly by the ocean**

Thank you for this comment, which is in line with Comment 8 from Reviewer 1. The phrase 'climatically vulnerable' has been removed from the text.

**16. L42 – Suggest you remove this –the case is convincing for ocean versus surface melting.**

As described in our response to the **Major Change 1**, we have revised the manuscript to provide a more balanced discussion of the role of both surface and ocean forcing. This clause has been edited to reflect this. New sentence reads:

"We then evaluate the potential mechanisms responsible for driving the observed seasonal ice-flow signals upstream of George VI Ice Shelf".

**17. L52 – very nice map, thank you**

**Thank you.**

18. L67 – Amplification is not the right word. Rapid increase.

Done.

19. L77 – unusual word choice. Mimicking.

**The word 'aping' has been replaced with 'similar to'.**

20. L82 – Add text along these lines: However, the George VI also has a long record of intense surface melting and surface ponding of melt, primarily in the central sector. Melt season duration is one of the longest on the continent at XXX (ref) and the adjacent Wilkins Ice Shelf has experienced episodes of hydrofracture-driven retreat (Braun Humbert or Scambos ref). Both processes will be examined as possible causes of the seasonal speedup.

**Thank you for this suggestion. We have assimilated most of these points into a now reworked Lines 61-72 where we think it is better placed given the surface forcing context. Lines read:**

"... These events have been attributed primarily to the surface warming-induced presence of supraglacial meltwater lakes (Dirscherl et al., 2021), which are surmised to have instigated a process of rapid ice-shelf hydrofracture and collapse such as that observed most recently at Wilkins Ice Shelf (Fig. 1; cf. Scambos et al., 2000; 2009; Banwell et al., 2013; Leeson et al., 2020). These phenomena have, in turn, been linked to an Antarctic Peninsula-wide increase in surface temperatures over most of the observational record (Vaughan et al., 2003). At GVIIS, intense supraglacial meltwater presence has been observed over the ice shelf since routine satellite observations began (Kingslake et al., 2017; Bell et al., 2018; Banwell et al., 2021), and melt season duration is one of the longest on the continent. This has led to the identification of GVIIS as a potential site for future ice-shelf disintegration (Holt and Glasser, 2022); recent modelling studies suggest that resultant land ice losses associated with such an event would contribute an ~8 mm rise in global sea-level by 2100 (Schannwell et al., 2018)."

**21. L92 – both ICESat-2 and MOA have mapped the grounding line – here is the new reference: Li, T., Dawson, G.J., Chuter, S.J. and Bamber, J.L. 2022. A high-resolution Antarctic grounding zone product from ICESat-2 laser altimetry. Earth System Science Data, 14(2), pp.535-557.**

We thank the reviewer for pointing us to this new dataset and that of MOA. Ultimately, the point we were trying to convey was the need for systematic, spatially continuous, and high-precision observations of the grounding line (or more technically: the limit of tidal flexure), since this location is vital for ensuring velocity signals do not fall either within or seaward of the grounding zone (see Rott et al. (2020) and references therein for the main motivation here). As the reviewer is aware, the Li et al. (2022) dataset relies upon (spatially discontinuous) repeat-track techniques, whereas MOA is limited by its moderate resolution compared to SAR imaging and its inability to reliably image the absolute limit of tidal flexure (although the authors nonetheless appreciate the value of this extremely useful dataset for other applications). As such, we have slightly rephrased this sentence for clarity:

"... In comparison to the detailed knowledge of grounding-line migration within these sectors, however, there have been no high-resolution, spatially complete grounding line surveys at GVIIS since the mid-1990s (Rignot et al., 2016). Accurate and updated knowledge of GVIIS' grounding line is therefore of paramount importance for distinguishing precisely between grounded and floating ice."

22. L95 – Remove after ice, -- the last part of this sentence should be in the intro of not used here

Thank you for this suggestion. We have removed the last clause of the sentence.

23. L112 – See the MAO data sets available from NSIDC, for 2004-5 and 2008-9 and I think there's a grounding line for 2014-15.

Thanks. See our response to Comment 21 above.

24. L136 - ... in one line of sight direction only.

Good spot. We have rephrased this sentence for clarity as recommended here and by Reviewer 1. Please see our response to Reviewer 1's Comment 20 for more detail.

25. L166 – outside of ? standard error bounds

We would prefer to keep the text as written. Pixels falling *within* standard error bounds were discarded because, given an error of  $\pm 1.8$  m yr-1, a seasonal signal between -1.8 m yr-1 and 1.8 m yr-1 cannot be trusted to be a true signal.

**26. Line 222 – no comma**

Thank you. This comma has been removed from the text.

**27. In this area, the glaciers are talking to one another.**

Thanks, this is an astute and interesting observation. As detailed in our response to Comment 5, we now discuss this phenomena in Section 4.2.2.

**28. Fig.6 - Consider moving this figure to supplemental information. It's a very nice confirmation but mostly in the direction of data quality and data issues.**

Following the clarifications and edits made in response to Reviewer 1 regarding this figure (Reviewer 1, Comments 30-32) and its description in Section 4.2.3, we would prefer to retain it as a main figure (Fig. 7 in the revised manuscript). We believe it serves as important confirmation on the reliability of our observed seasonal signal, and further highlights (in addition to Fig. 3) the spatial distribution of the speedup along the coast.

- 29. L316 ...but they are persistent in the areas that they occur. It's not 'rare' in areas where there is a significant melt season.
- 30. L318 Well, yes no one has done as thorough an analysis as you. How would they ever be identified if precedent (not seen before...) is the justification for not considering the possibility?
- 31. L321 Banwell notes abundant melt ponds on the GVIIS in several years. It is wrong to imply that there is not significant surface melting in this grounded ice area! GVIIS has one of the longest melt seasons on the continent
- 32. L324 This is ad hoc, -you- are making the best case in this work, this is how one discovers that water is involved in grounded-ice seasonal variations.

Thank you for these comments. As stipulated in **Major Change 1**, we have overhauled the discussion to present a more balanced evaluation of the relative roles of surface meltwater and ocean forcing, which we trust directly addresses each of these.

**33. L327 – these papers say nothing about a strong seasonality in flow speed or CDW depth.**

We agree that neither Jenkins and Jacobs (2008) nor Meredith et al. (2010) reference any seasonality with regards to CDW flow speed or depth. Discussion of these papers in the original text was intended merely to first highlight the existence of CDW in the cavity beneath GVIIS.

In our revised Discussion (cf. **Major Change 1**) we have now stipulated this point for clarity (sentence opening the second paragraph of text in Section 5.2). This addition also serves to better allude to the following series of arguments on the mechanisms through which seasonal CDW influx variability may occur (see also our response to Comment 3 and 35).

**Revised lines read:**

"It is important to note, however, that the findings of Jenkins and Jacobs (2008) do not present any evidence for seasonality in CDW presence and/or depth owing to the limited timeframe in which their in situ observations were collected (less than two days' worth of continuous measurements in March

1994). Nonetheless, recent research has revealed two possible mechanisms through which sea ice conditions offshore from GVIIS may control CDW draft in its sub-shelf cavity. First, ...".

**34. L333 – The JandJ 2008 paper indicates flow speeds of ~2cm/sec. A monthly transit of the 500 km-long GVIIS cavity –which is what you are implying to create the narrow summer speed-up –would require speed near 20 cm/s, and that is if the water moved laminarly and linearly.**

This is an interesting thought exercise which we now initially elaborate upon in the revised discussion as a potential ocean forcing 'con' (Revised Section 5.2). (Note here that we have re-performed the calculation using more precise measurements of the cavity's long axis (420 km) and a presumed transit time ranging between 1-2 months which is potentially more realistic of the duration needed to yield the speedup trends observed – this gives a speed ranging between ~8-16 cm s-1).

In the context of our new, wider discussion on a lack of ocean observational constraints, we also compare the 2 cm s-1 value estimated by Jenkins and Jacobs (2008) (which was estimated from only ~30 hours of observations in 1994) to values observed elsewhere in Antarctica over much longer timescales. These observations reveal speeds of up to 8-20 cm s-1 (see Jacobs et al., 2013; Jenkins et al., 2018), suggesting that our calculated GVIIS rate of 8-16 cm/s may not be entirely unreasonable. As such, this finding is also presented as a 'pro' in favour of ocean forcing, all the while providing the motivation/need for much more ocean research in this area of the world.

**This new section reads:**

"Ultimately, a historical dearth of oceanographic observations in the Bellingshausen Sea hinders our ability to ascertain which mechanism is the dominant control on CDW influx to GVIIS' sub-shelf cavity, justifying the future collection of detailed oceanographic data in this region. Such data would also yield high-resolution (and potentially more representative) insights into the nature of oceanic circulation beneath GVIIS. Indeed, we estimate that an ~8-16 cm s-1 northwards throughflow of CDW would be required over the course of ~1-2 months to induce the relatively narrow summertime speedup windows observed along the entirety of GVIIS' 420 km long cavity (Figs. 4, 5 and A1), assuming laminar and linear flow. These rates are up to almost an order of magnitude greater than the observationally constrained estimates reported in Jenkins & Jacobs (2008; ~2.5 cm s-1), although the latter, which were collected over ~30 hours, may not necessarily be representative of seasonally averaged rates of flow. Elsewhere in Antarctica, longer-term in situ observations have revealed much greater rates of sub-ice shelf CDW circulation (~8-20 cm s-1; Jacobs et al., 2013; Jenkins et al., 2018), suggesting that similar speeds may, in fact, be plausible underneath GVIIS."

**35. L336 – I am suggesting a complete re-write of this section, and the summary.**

We presume the reviewer's motivation for this comment arises from Comments 3 and 9, which make reference to a lack of any discussion pertaining to the mechanism by which seasonality in CDW influx may occur (which we interpret to be a possible oversight on the part of the reviewer given Lines 336-356 of our initial manuscript).

Nonetheless, we have overhauled the discussion as advised (see also our responses to **Major Change 1** and Comments 2-4 and 33-34) which we hope will address the reviewer's concerns. Similar to the abstract, we have also reworded the summary to offer a more balanced conclusion on the relative roles of both surface and ocean forcing.

**36. Figure A1:**

- Flowline 3 this signal, and many of the others, is inconsistent with ocean forcing for the seasonal speed-up.
- Flowline 5 compare this pattern with flowline 20: again, opposite seasonal phasing
- Flowline 20 [compare] with flowline 5, Ryder
- Flowline 21 This glacier is on the opposite side of the GVI, with an opposite pattern of seasonal variation.

We thank the reviewer for his insightful comments on this figure. These have now been addressed in response to Comments 3 and 5 above.

**References cited in this rebuttal letter**

- Banwell, A.F., Datta, R.T., Dell, R.L., Moussavi, M., Brucker, L., Picard, G., Shuman, C.A., Stevens, L.A., 2021. The 32-year record-high surface melt in 2019/2020 on the northern George VI Ice Shelf, Antarctic Peninsula. *The Cryosphere* 15, 909–925. https://doi.org/10.5194/tc-15-909-2021
- Bell, R.E., Banwell, A.F., Trusel, L.D. and Kingslake, J., 2018. Antarctic surface hydrology and impacts on ice-sheet mass balance. Nature Climate Change, 8(12), pp.1044-1052. https://doi.org/10.1038/s41558-018-0326-3
- Boxall, K., Christie, F.D.W., Willis, I.C., Wuite, J. And Nagler, T. 2022a. West Antarctic Peninsula grounding line location datasets supporting "Seasonal land ice-flow variability in the Antarctic Peninsula". [Dataset] https://doi.org/10.17863/CAM.82248
- Boxall, K., Christie, F.D.W., Willis, I.C., Wuite, J. And Nagler, T. 2022b. West Antarctic Peninsula seasonal ice velocity products supporting "Seasonal land ice-flow variability in the Antarctic Peninsula". [Dataset] https://doi.org/10.17863/CAM.82252
- Carrasco, J.F., Bozkurt, D. and Cordero, R.R., 2021. A review of the observed air temperature in the Antarctic Peninsula. Did the warming trend come back after the early 21st hiatus?. *Polar Science*, 28, p.100653. https://doi.org/10.1016/j.polar.2021.100653
- Christie, F.D.W., Benham, T.J., Batchelor, C.L., Rack, W., Montelli, A. and Dowdeswell, J.A., 2022. Antarctic ice-shelf advance driven by anomalous atmospheric and sea-ice circulation. Nature Geoscience, 15(5), pp.356-362. https://doi.org/10.1038/s41561-022-00938-x
- Friedl, P., Seehaus, T. and Braun, M., 2021. Global time series and temporal mosaics of glacier surface velocities derived from Sentinel-1 data. *Earth System Science Data*, 13(10), pp.4653-4675. https://doi.org/10.5194/essd-13-4653-2021
- Greene, C.A., Young, D.A., Gwyther, D.E., Galton-Fenzi, B.K. and Blankenship, D.D., 2018. Seasonal dynamics of Totten Ice Shelf controlled by sea ice buttressing. The Cryosphere, 12(9), pp.2869-2882. https://doi.org/10.5194/tc-12-2869-2018
- Hogg, A.E., Shepherd, A., Cornford, S.L., Briggs, K.H., Gourmelen, N., Graham, J.A., Joughin, I., Mouginot, J., Nagler, T., Payne, A.J., Rignot, E., Wuite, J., 2017. Increased ice flow in Western Palmer Land linked to ocean melting. Geophys. Res. Lett. 44, 4159–4167. https://doi.org/10.1002/2016GL072110
- Holland, P. R., A. Jenkins, and D. M. Holland, 2010, Ice and ocean processes in the Bellingshausen Sea, Antarctica, J. Geophys. Res., 115, C05020, https://doi.org/10.1029/2008JC005219
- Holt, T.O., Glasser, N.F., Quincey, D.J. and Siegfried, M.R., 2013. Speedup and fracturing of George VI Ice Shelf, Antarctic Peninsula. The Cryosphere, 7(3), 797-816. https://doi.org/10.5194/tcd-7-373-2013
- Holt, T. and Glasser, N.F., 2022. Changes in area, flow speed and structure of southwest Antarctic Peninsula ice shelves in the 21st century. *Journal of Glaciology*, pp.1-19. https://doi.org/10.1017/jog.2022.7
- Howat, I.M., Porter, C., Smith, B.E., Noh, M.-J., and Morin, P., 2019. The Reference Elevation Model of Antarctica. *The Cryosphere 13*, 665–674. https://doi.org/10.5194/tc-13-665-2019

- Jacobs, S., Giulivi, C., Dutrieux, P., Rignot, E., Nitsche, F., and Mouginot, J., 2013. Getz Ice Shelf melting response to changes in ocean forcing. *J. Geophys. Res.*,118, 4152-4168. doi:10.1002/jgrc.20298
- Jenkins, A., and S. Jacobs, 2008, Circulation and melting beneath George VI Ice Shelf, Antarctica, J. Geophys. Res.,113, C04013, https://doi.org/10.1029/2007JC004449
- Jenkins, A., Shoosmith, D., Dutrieux, P., Jacobs, S., Kim, T.W., Lee, S.H., Ha, H.K. and Stammerjohn, S., 2018. West Antarctic Ice Sheet retreat in the Amundsen Sea driven by decadal oceanic variability. *Nature Geoscience*, 11(10), pp.733-738. https://doi.org/10.1038/s41561-018-0207-4
- Johnson, A., Hock, R. and Fahnestock, M., 2022. Spatial variability and regional trends of Antarctic ice shelf surface melt duration over 1979–2020 derived from passive microwave data. Journal of Glaciology, 68(269), pp.533-546. https://doi.org/10.1017/jog.2021.112
- Li, T., Dawson, G.J., Chuter, S.J. and Bamber, J.L., 2022. A high-resolution Antarctic grounding zone product from ICESat-2 laser altimetry. Earth System Science Data, 14(2), pp.535-557. https://doi.org/10.5194/essd-14-535-2022
- Meredith, M.P., Wallace, M.I., Stammerjohn, S.E., Renfrew, I.A., Clarke, A., Venables, H.J., Shoosmith, D.R., Souster, T., Leng, M.J., 2010. Changes in the freshwater composition of the upper ocean west of the Antarctic Peninsula during the first decade of the 21st century. Prog. Oceanogr., 3rd GLOBEC OSM: From ecosystem function to ecosystem prediction 87, 127–143. https://doi.org/10.1016/j.pocean.2010.09.019
- Moon, T., Joughin, I., Smith, B., Van Den Broeke, M.R., Van De Berg, W.J., Noël, B., Usher, M., 2014. Distinct patterns of seasonal Greenland glacier velocity. *Geophys. Res. Lett.* 41, 7209–7216. https://doi.org/10.1002/2014GL061836
- Rack, W., King, M. A., Marsh, O. J., Wild, C. T. & Floricioiu, D., 2017. Analysis of ice shelf flexure and its InSAR representation in the grounding zone of the southern McMurdo Ice Shelf. *Cryosphere* 11(6), 2481–2490. doi:10.5194/tc-11-2481-2017.
- Rignot, E., 2016. MEaSUREs Antarctic Grounding Line from Differential Satellite Radar Interferometry, Version 2. https://doi.org/10.5067/IKBWW4RYHF1Q
- Rott, H., Wuite, J., De Rydt, J., Gudmundsson, G.H., Floricioiu, D. and Rack, W., 2020. Impact of marine processes on flow dynamics of northern Antarctic Peninsula outlet glaciers. *Nature communications*, 11(1), 1-3. https://doi.org/10.1038/s41467-020-16658-y
- Scambos, T., Fricker, H.A., Liu, C.C., Bohlander, J., Fastook, J., Sargent, A., Massom, R. and Wu, A.M., 2009. Ice shelf disintegration by plate bending and hydro-fracture: Satellite observations and model results of the 2008 Wilkins ice shelf break-ups. *Earth and Planetary Science Letters*, 280(1-4), pp.51-60. https://doi.org/10.1016/j.epsl.2008.12.027
- Seehaus, T., Marinsek, S., Helm, V., Skvarca, P. and Braun, M., 2015. Changes in ice dynamics, elevation and mass discharge of Dinsmoor–Bombardier–Edgeworth glacier system, Antarctic Peninsula. *Earth and Planetary Science Letters*, 427, pp.125-135. https://doi.org/10.1016/j.epsl.2015.06.047
- Seehaus, T.C., Marinsek, S., Skvarca, P., Van Wessem, J.M., Reijmer, C.H., Seco, J.L. and Braun, M.H., 2016. Dynamic response of sjögren inlet glaciers, antarctic peninsula, to ice shelf breakup

derived from multi-mission remote sensing time series. *Frontiers in Earth Science*, 4, p.66. https://doi.org/10.3389/feart.2016.00066

- Trusel, L.D., Frey, K.E., Das, S.B., Munneke, P.K., Broeke, M.R. van den, 2013. Satellite-based estimates of Antarctic surface meltwater fluxes. Geophys. Res. Lett. 40, 6148–6153. https://doi.org/10.1002/2013GL058138
- Vijay, S. and Braun, M., 2017. Seasonal and interannual variability of Columbia Glacier, Alaska (2011–2016): Ice velocity, mass flux, surface elevation and front position. *Remote Sensing*, 9(6), p.635. https://doi.org/10.3390/rs9060635
- Webber, B.G., Heywood, K.J., Stevens, D.P., Dutrieux, P., Abrahamsen, E.P., Jenkins, A., Jacobs, S.S.,
  Ha, H.K., Lee, S.H. and Kim, T.W., 2017. Mechanisms driving variability in the ocean forcing of
  Pine Island Glacier. *Nature communications*, 8(1), pp.1-8.
  https://doi.org/10.1038/ncomms14507

---

## Referee Report (RR1)

Re-review notes for Boxall et al.,

**Seasonal land ice-flow variability in the Antarctic Peninsula (tc-2022-55)**

A few comments –
The discussion in 5.1 is now very good, fundamentally what is needed are observations of melt days at a smaller spatial resolution than are currently widely available. Yes, the gradient of melt days with altitude is steep, but there are melt ponds at a few hundred meters' elevation in the AP, on the eastern side (where they are revealed more easily due to low snow accumulation). See attached photo.

No real changes to suggest to Section 5.1 except to remove the last sentence in the secton: *We would expect any subsurface meltwater to perpetually flow towards the grounding line due to gravity.* It gives the wrong impression, that a firn aquifer on a slope would 'drain away' and not accumulate water. As Harper et al., 2012 and Koenig et al., 2013 note, there are perennial aquifers on the grounded and sloping Greenland ice sheet, and they pool water in the topographic lows and spill over, slowly and perennially, toward the coast.

Ok, one more comment: the cooling trend is real (and likely tied to shifts in the PDO), but it's not like surface melting stopped anywhere where it was common before. And the cooling trend, following the PDO shift again, may be reversing. As you note, Banwell et al., 2021, showed near-record melting in 2020-2021 in the region.

On Section 5.2, I note that while the opening paragraph makes a strong case for influx of CDW, and I'm sure that is happening, there is not a strong case for seasonality of the inflow. Ah, I see that is the first line of the next paragraph.

In the first paragraph, for this last sentence: *with inferred patterns of melting observed recently along the Coriolis- favoured flank of Dotson Ice Shelf, West Antarctica (Gourmelen et al., 2017).* You might also cite the more wide-ranging work of Karen Alley (note, I'm a co-author on these papers, but they do cover many more of these features than the excellent Gourmelen work).

Paragraph beginning 'Ultimately', I think you don't need the word 'historical' in that sentence.

Reviewer #1 response, Comment #4: The near-grounding line focus of the speed-up, no matter the cause (ocean or surface melt) is more likely an indication that basal shear stress in the glaciers is large, preventing a rapid or extensive transfer of reduced longitudinal compression upstream. This is true for several coastal areas of Antarctica (e.g., Getz Ice Shelf, which -is- seeing a strong ocean-derived basal melting, but little upstream propagation of increased flow speed).

I read through all the responses to the Reviewer 2 comments (mine) and I am fine with all of them.

My recommendation to the editor is that the paper is ready to publish from a science standpoint.

[Figure]

A picture of Crane Glacier, looking downstream; the large blue area is an ice-capped meltwater lake. The image was taken in April 2013. The point being that extensive melting does occur upstream of the grounding line in the AP; for the glaciers feeding the GVIIS, a bit more spatial resolution in melt-day mapping is required (for another study at another time).

---

## Referee Report (RR2)

The quality of the manuscript strongly improved and most issues were addressed adequately. In particular the discussion improved strongly and provides a very nice and interesting summary of the potential processes explaining the observed velocity patterns, highlights the needs for future research. Very good job!

There are only very fee issues open, which should be addressed before publication of the paper.

Most important is the poor data coverage for some flow lines and a clear description of the error computation.

Detailed comments:

- L280: Maybe you should note the poor data coverage at flow line 10. This might partly explain its pattern.

- Fig. 4 and 5: Moreover, it is still unclear, if you analyze only time periods with a "13-month Moving Mean" available in Fig. 4 to generate Fig. 5. In particular for flow line 16 the coverage by the "13-month Moving Mean" is quite small. Well, flow lines 5, 9, 15 and 17 have also quite limited temporal coverage. At least, you should mention this issue and its implications for Fig. 5.
- Moreover, it is still unclear how you computed the error bars. In line 127ff, you explain the calculation for the mosaics, but it is unclear how you obtained the error for the 10km² regions (mean of SE within the 10km² areas?) and for the detrended monthly ice flow (Fig.5, SE of all individual vel. Fields for a certain month?, how did you compute the error throughout the 10km² areas?). You talk about standard error in the figure caption and in the legend it is called mean standard error. Please clarify.

- L300 ff. This is an interesting hypothesis but quite speculative. What about flowline17-19 vs 6-9? You should rephrase it. e.g. remove "clear evidence" and state that more research is needed. I would suggest to analyze this hypothesis on yearly scales and based on long-term average.

- Section D1: I am still a bit confused by the plotted "fitted cosine wave". Flow line 6: Dominant frequency=3. As far as I understand your analysis a frequency of "1" stands for one full cycle per year. I would expect, that a frequency of 3 relates to "3" full cycles per year. However, the orange line for flow line 6 shows only one full cycle per year. Same for flow line 8. What is the unit of phase is it in "rad" or in "month"?
What is the unit of the amplitude?

---

## Author Response (AR2)

**Seasonal land ice-flow variability in the Antarctic Peninsula (tc-2022-55)**

**Author Response to Reviewers (20/08/22)**

Dear Dr. Berthier,

We thank both reviewers for their comments on our revised manuscript and were pleased to read that they find it to be "*very good"* (Reviewer 2) and that "*the quality of the manuscript strongly improved*" (Reviewer 1). We were particularly pleased to read that Reviewer 2, Dr. Ted Scambos, recommends the manuscript "*is ready to publish from a science standpoint*".

In the following response document, we provide point-by-point responses to the reviewers' final minor comments. As before, we have included the numbered reviewers' comments (*italicised in blue*), our responses (black text) and amendments to the original text (*italicised in grey*). Unless otherwise stated, line and figure numbers referred to below are in line with the revised manuscript with tracked changes submitted here.

Following your instructions, we have also checked carefully the manuscript for typos, missing co-authors and their affiliations, terminology, updates of data in tables, or updates of variables in equations and confirm that there are none throughout.

Kind regards,

Karla Boxall

(on behalf of the co-author team)

**Reviewer #1**

1. *The quality of the manuscript strongly improved and most issues were addressed adequately. In particular the discussion improved strongly and provides a very nice and interesting summary of the potential processes explaining the observed velocity patterns, highlights the needs for future research. Very good job!*

We thank the reviewer for their positive re-review, and were pleased to read they believe the manuscript has "strongly improved" and that the revised Discussion (Section 5) "provides a very nice and interesting summary of the potential processes explaining the observed velocity patterns".

2. *There are only very fee* [few] *issues open, which should be addressed before publication of the paper.*
3. *Most important is the poor data coverage for some flow lines and a clear description of the error computation.*

We thank the reviewer for raising these points. Please see our responses to the detailed comments below regarding the poor temporal data coverage for some flowlines, and the description of our standard error calculation.

*Detailed comments:*

4. *L280: Maybe you should note the poor data coverage at flow line 10. This might partly explain its pattern.*

We have now noted the poor temporal data coverage at flowline 10 as a potential reason for the observed non-summertime speedup signal. The revised text on Line 270 now reads:

*"The latter likely arises from poor temporal data coverage, while the double peak associated with Flowline 8 is attributed to an anomalous wintertime speedup in 2016 which dominated the mean velocity signal of this glacier."*

5. *Fig. 4 and 5: Moreover, it is still unclear, if you analyze only time periods with a "13-month Moving Mean" available in Fig. 4 to generate Fig. 5. In particular for flow line 16 the coverage by the "13-month Moving Mean" is quite small. Well, flow lines 5, 9, 15 and 17 have also quite limited temporal coverage. At least, you should mention this issue and its implications for Fig. 5.*

Thank you for this comment. We have reworded the text on Line 176 to clarify that only the portions of each velocity time series with 13 months of continuous data were detrended.

*"Next, to accentuate intra-annual trends, we removed long-term trends in velocity over each 10 km² region (cf. Hogg et al., 2017; Gardner et al., 2018) using a 13-month moving average. At flowlines where the removal of a 13-month moving average was not possible, for example due to gaps in the time series, the data were excluded from our subsequent calculation of detrended monthly means."*

6. *Moreover, it is still unclear how you computed the error bars. In line 127ff, you explain the calculation for the mosaics, but it is unclear how you obtained the error for the 10km² regions (mean of SE within the 10km² areas?) and for the detrended monthly ice flow (Fig.5, SE of all individual vel. Fields for a certain month?, how did you compute the error throughout the 10km² areas?). You talk about standard error in the figure caption and in the legend it is called mean standard error. Please clarify.*

We have added the word 'mean' to Line 173 to clarify that for each 10 km$^2$ region, we calculated a mean velocity and a mean standard error, for each month. The text now reads:

*"To do this, we calculated mean velocity and mean standard error (cf. Sect. 3.2) at monthly intervals within a 10 km$^2$ region located directly upstream of the grounding line between 2014 and 2020."*

Both the legend and the figure caption of Fig. 5 have been updated to read "mean monthly standard error" to clarify that the grey shading in Fig. 5 represents the mean standard error over each 10 km$^2$ region, averaged by month.

> 7. *L300 ff. This is an interesting hypothesis but quite speculative. What about flowline17-19 vs 6-9? You should rephrase it. e.g. remove "clear evidence" and state that more research is needed. I would suggest to analyze this hypothesis on yearly scales and based on long-term average.*

We thank the reviewer for their comments on this section, which was of course added at the request of Reviewer 2 (Dr. Ted Scambos). We appreciate that these signals are not universally observed across GVIIS' outlet glaciers and, in accordance with the reviewer's advice, have therefore slightly reworded the sentence as follows. Similarly, we have also added a new sentence on Line 296 to stipulate that more research and observations are ultimately required. Revised text reads:

*"Within this trend, Figs. 4 and 5 also present evidence of the ability of select GVIIS' outlet glaciers to influence each other across the ice shelf, whereby the earlier acceleration of Alexander Island's glaciers initially arrest those flowing from Palmer Land, delaying the onset of their acceleration until the late summertime (compare, for example, the timing of peak velocity at the geographically opposite Flowlines 21 and 3, and Flowlines 20 and 4-5). Upon late-summertime acceleration, Palmer Land's glaciers then arrest the flow of those on Alexander Island in a similar manner. Continued monitoring of this phenomenon beyond the current Sentinel-1a/b observational record may shed additional light on the importance of this mechanism (or otherwise) for controlling the overall seasonal signals exhibited at these outlet glaciers."*

> 8. *Section D1: I am still a bit confused by the plotted "fitted cosine wave". Flow line 6: Dominant frequency=3. As far as I understand your analysis a frequency of "1" stands for one full cycle per year. I would expect, that a frequency of 3 relates to "3" full cycles per year. However, the orange line for flow line 6 shows only one full cycle per year. Same for flow line 8. What is the unit of phase is it in "rad" or in "month"? What is the unit of the amplitude?*

For consistency in the interpretation of the amplitude and phase values, we chose to fit cosine waves with a constant frequency to all velocity time series. This constant frequency was ascertained from the most dominant frequency observed across all timeseries.

This is stated in the text in Appendix D:

*"… a cosine wave, optimised to the most dominant frequency observed across all outlet glacier time series (1 month), is fitted to each time series"*

It is also included in the caption of Fig. D1:

*"Panels show mean monthly ice flow of each outlet glacier (blue; same as Fig. 5) with fitted cosine functions (orange) optimised to the most dominant frequency observed across all outlet glacier time series."*

The unit of phase is radians, and the unit of amplitude is m d$^{-1}$. These units have been added to the headings in Table D1 and to the axis label in Figure D1.

1. *A few comments – The discussion in 5.1 is now very good, fundamentally what is needed are observations of melt days at a smaller spatial resolution than are currently widely available. Yes, the gradient of melt days with altitude is steep, but there are melt ponds at a few hundred meters' elevation in the AP, on the eastern side (where they are revealed more easily due to low snow accumulation). See attached photo.*

We thank the reviewer, Dr. Ted Scambos, for his re-review and were pleased to read he is contented with our revised Discussion (Section 5.1). We agree that currently there is not enough observationally constrained evidence at the required spatial resolution to implicate surface forcing as the primary mechanism driving the observed seasonality.

2. *No real changes to suggest to Section 5.1 except to remove the last sentence in the section: We would expect any subsurface meltwater to perpetually flow towards the grounding line due to gravity. It gives the wrong impression, that a firn aquifer on a slope would 'drain away' and not accumulate water. As Harper et al., 2012 and Koenig et al., 2013 note, there are perennial aquifers on the grounded and sloping Greenland ice sheet, and they pool water in the topographic lows and spill over, slowly and perennially, toward the coast.*

Thank you for this comment. We have removed the sentence from Line 384 as suggested.

3. *Ok, one more comment: the cooling trend is real (and likely tied to shifts in the PDO), but it's not like surface melting stopped anywhere where it was common before. And the cooling trend, following the PDO shift again, may be reversing. As you note, Banwell et al., 2021, showed near-record melting in 2020-2021 in the region.*

In response to this comment, we have reworded Line 376 to tone down the assertion that the cooling trend observed over the Antarctic Peninsula since the late 1990s by Turner et al. (2016) renders the existence of firn aquifers improbable. The sentence now reads:

*"We note, however, that the formation and persistence of firn aquifers requires high levels of surface melt and accumulation (Harper et al., 2012; Koenig et al., 2012; Montgomery et al., 2020): phenomena which may not be prevalent inland of GVIIS during most of the Sentinel-1 era given the pervasive cooling of the Antarctic Peninsula over approximately the past two decades noted above (cf. Turner et al., 2016)."*

4. *On Section 5.2, I note that while the opening paragraph makes a strong case for influx of CDW, and I'm sure that is happening, there is not a strong case for seasonality of the inflow. Ah, I see that is the first line of the next paragraph.*

We agree that Jenkins and Jacobs (2008) do not reference any seasonality with regards to CDW influx. We are glad to read that our first sentence in the second paragraph makes this point clear.

5. *In the first paragraph, for this last sentence: with inferred patterns of melting observed recently along the Coriolis- favoured flank of Dotson Ice Shelf, West Antarctica (Gourmelen et al., 2017). You might also cite the more wide-ranging work of Karen Alley (note, I'm a co-author on these papers, but they do cover many more of these features than the excellent Gourmelen work).*

We thank the reviewer for this helpful suggestion. We believe Karen et al. (2016) is particularly relevant here. This citation has been added to Line 399.

6. *Paragraph beginning 'Ultimately', I think you don't need the word 'historical' in that sentence.*

Thanks. We have removed the word 'historical' from Line 416.

7. *Reviewer #1 response, Comment #4: The near-grounding line focus of the speed-up, no matter the cause (ocean or surface melt) is more likely an indication that basal shear stress in the glaciers is large, preventing a rapid or extensive transfer of reduced longitudinal compression upstream. This is true for several coastal areas of Antarctica (e.g., Getz Ice Shelf, which –is seeing a strong ocean-derived basal melting, but little upstream propagation of increased flow speed).*

We thank the reviewer for sharing this insight. While no action is required for the current paper, this is a useful concept which we will keep in mind for our future research.

8. *I read through all the responses to the Reviewer 2 comments (mine) and I am fine with all of them.*
9. *My recommendation to the editor is that the paper is ready to publish from a science standpoint.*

Thank you. These statements are pleasing to read.

10. *A picture of Crane Glacier, looking downstream; the large blue area is an ice-capped meltwater lake. The image was taken in April 2013. The point being that extensive melting does occur upstream of the grounding line in the AP; for the glaciers feeding the GVIIS, a bit more spatial resolution in melt-day mapping is required (for another study at another time).*

We thank the reviewer for sharing this image. This will be of good use in our future, follow-up research.

---

## Author Response (AR3)

**Seasonal land ice-flow variability in the Antarctic Peninsula (tc-2022-55)**

**Author Response to Reviewers (30/08/22)**

1. *L27. "Since 1992". I suggest rather providing the start and end date of the IMBIE estimate. The paper being from 2018 I assume the end date of this estimate is not "now" (as understated while using "since") but sometime in 2015 or 2016 (or 2017?). An end date is thus needed.*

Sentence now reads:

*'... from which resulting land ice losses are estimated to have totalled an average of ~109 ± 59 gigatons per year between 1992 and 2017 (The IMBIE Team, 2018).'*

2. *L30. Authors do not need to include "cf." before a reference. It is implicit (to be removed elsewhere also, several occurrences).*

Instances of 'cf.' have been removed throughout the text.

3. *L131. Calculation of the standard error in equation (1). Dividing by the square root of the number of pixels implies independence (i.e. no spatial correlation) of neighbouring pixels. Do the authors have a justification for this choice? For example, in the field of DEM differencing, accounting for spatial correlation is key (and now standard) to reach a proper uncertainty estimate, otherwise the standard error is way too small (Rolstad, C., Haug, T., and Denby, B.: Spatially integrated geodetic glacier mass balance and its uncertainty based on geostatistical analysis: application to the western Svartisen ice cap, Norway, J Glaciol, 55, 666–680, 2009.)*

The calculation of the standard error was carried out on a per-pixel basis. We divided by the square root of the valid pixel count, which is the number of non-NaN observations used in the production of each monthly estimate, rather than dividing by the number of pixels. The text has been edited to clarify this point, and now reads:

*'... the mean per-pixel standard error totals 0.005 m d$^{-1}$ (1.8 m yr$^{-1}$). This value is comparable to that of other SAR-derived velocity products (Rignot et al., 2017; Friedl et al., 2021), and was calculated for each pixel...'*

4. *L166. A dot is missing after the parenthesis.*

A full stop has been added to the end of this sentence.

5. *L169. Should not the authors state "Pixels for which velocity was falling within..."?*

The sentence has been rephrased to read:

*'Pixels where the velocity fell within standard error bounds (Sect. 3.2) were also discarded.'*

6. *L514. I tried to access your data (26 August) and the doi was not found. Maybe the repository will be opened later, at the time of final publication. To be double checked.*

The link to the data in the repository will be minted upon receipt of the DOI of the accepted paper.